# MLR-Bench: Evaluating AI Agents on Open-Ended Machine Learning Research

**Hui Chen**[◇][*]   **Miao Xiong**[◇][*]   **Yujie Lu**[♡]   **Wei Han**[♠]   **Ailin Deng**[◇]

**Yufei He**[◇]   **Jiaying Wu**[◇]   **Yibo Li**[◇]   **Yue Liu**[◇]   **Bryan Hooi**[◇]

◇National University of Singapore    ♡ University of California, Santa Barbara
♠ Singapore University of Technology and Design

{hui.chen, dcsbhk}@nus.edu.sg, miao.xiong@u.nus.edu

⌂: https://github.com/chchenhui/mlrbench

## Abstract

Recent advancements in AI agents have demonstrated their growing potential to drive and support scientific discovery. In this work, we introduce **MLR-Bench**, a comprehensive benchmark for evaluating AI agents on open-ended machine learning research. MLR-Bench includes three key components: (1) 201 research tasks sourced from NeurIPS, ICLR, and ICML workshops covering diverse ML topics; (2) **MLR-Judge**, an automated evaluation framework combining LLM-based reviewers with carefully designed review rubrics to assess research quality; and (3) **MLR-Agent**, a modular agent scaffold capable of completing research tasks through four stages: idea generation, proposal formulation, experimentation, and paper writing. Our framework supports both *stepwise* assessment across these distinct research stages, and *end-to-end* evaluation of the final research paper. We then use MLR-Bench to evaluate six frontier LLMs and an advanced coding agent, finding that while LLMs are effective at generating coherent ideas and well-structured papers, current coding agents frequently (e.g., in 80% of the cases) produce fabricated or invalidated experimental results—posing a major barrier to scientific reliability. We validate MLR-Judge through human evaluation, showing high agreement with expert reviewers, supporting its potential as a scalable tool for research evaluation. We open-source MLR-Bench to help the community benchmark, diagnose, and improve AI research agents toward trustworthy and transparent scientific discovery.

## 1   Introduction

A hallmark of human intelligence is the ability to generate new knowledge through research and scientific discovery. From basic research to applied technological development, scientific progress has shaped human civilization. Reproducing this ability—to autonomously generate, test, and validate new knowledge—has long been viewed as one of the grand challenges for artificial intelligence.

Recent advances in large language models (LLMs) bring us closer to this vision. LLM-powered agents have demonstrated impressive capabilities in generating research ideas [16, 3, 28, 17], conducting experiments and analyses [11, 34, 4, 40, 13, 41, 15], drafting scientific articles [25, 35, 38, 1] and automated review [36, 14, 32, 6, 42]. These advances move us beyond automating isolated tasks, opening up the possibility of automating the entire scientific process, from idea conception to experimentation to dissemination.

---

[*]Equal contribution

39th Conference on Neural Information Processing Systems (NeurIPS 2025) Track on Datasets and Benchmarks.

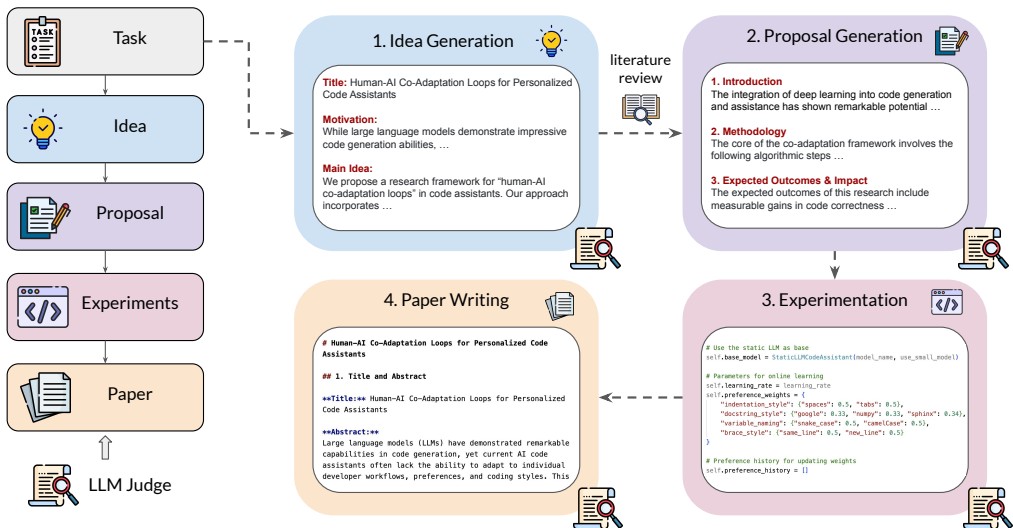

Figure 1: An overview of the framework of MLR-Bench, consisting of both an end-to-end evaluation (left) and a stepwise evaluation (right), each of which uses LLM judges to automatically assess performance over 201 tasks. For end-to-end evaluation, we use the same model as backbone in idea generation, proposal generation and paper writing. For stepwise evaluation, various models are tested and compared within each step.

However, realizing this vision raises an important question: *how do we rigorously evaluate the quality of research produced by AI agents?* While recent works have shown promising results [29, 27, 19, 26, 39], the community still lacks a comprehensive benchmark to systematically assess AI agents' ability to conduct open-ended scientific research, making progress toward autonomous scientific discovery difficult to measure and compare fairly between agents. Moreover, there is also a need for empirical analysis to identify key failure modes—such as hallucinated results, lack of novelty, or methodological flaws — to quantify current limitations and inform future progress in research agents.

Toward this vision, we introduce **MLR-Bench**, a comprehensive benchmark designed for evaluating AI agents on *open-ended machine learning research tasks*. Specifically, we seek to answer the following research questions:

- **RQ1:** *How well can AI agents conduct open-ended machine learning research?* (§3)

- **RQ2**: *How effectively can an LLM judge evaluate research, as measured by its agreement with human reviewers?* (§4)

- **RQ3:** *What are the key factors that affect the quality of AI-generated research?* (§5)

To systematically address these questions, we decompose the research process into four successive stages: (1) *idea generation*, (2) *research proposal formulation*, (3) *experimental execution and analysis*, and (4) *paper writing*. This decomposition allows us to assess both stepwise progress and the overall end-to-end research capability of AI agents.

MLR-Bench consists of three key components: (1) a collection of 201 real-world research tasks sourced from NeurIPS, ICLR, and ICML workshops over the past three years, covering a wide range of machine learning domains, including LLMs, trustworthy AI, ML systems, AI for science; (2) **MLR-Judge**, an automated evaluation pipeline consisting of LLM-based judge and structured review rubrics; and (3) **MLR-Agent**, a modular research agent scaffold capable of automatically completing these tasks by following the four defined research stages.

To explore RQ1, we use MLR-Bench to evaluate six state-of-the-art models—including the most recent o4-mini [24], Gemini-2.5-pro-preview [8], and Qwen3-235B-A22B [31]—as well as the advanced coding agents, Claude Code [2] and Codex [23]. Our results show that while these models perform well in generating ideas and writing papers, they often fail to deliver innovative

or scientifically reliable research, mostly due to coding agents producing invalidated experimental results.

To validate the reliability of MLR-Judge (addressing RQ2), we recruit 10 machine learning experts with top-tier conference review experience to independently review the AI-generated research outputs. We find that the level of agreement between an LLM judge and a human judge, when assessing the same output, is very close to the level of agreement between two human judges, demonstrating its effectiveness as a reliable evaluation tool. Finally, to address RQ3, we analyze the justifications provided by both MLR-Judge and human reviewers, identifying a recurring failure mode where agents tend to generate fabricated or unverified results after execution failures—revealing a gap between fluent output generation and true scientific rigor.

Our contributions include:

- **MLR-Bench**: To our knowledge, the most comprehensive evaluation benchmark for AI research agents to date, featuring 201 open-ended ML research tasks, a human-aligned **MLR-Judge** for automated research quality assessment, and a modular **MLR-Agent** supporting both stepwise and end-to-end research execution.
- **Comprehensive Evaluation and Insights**: We apply MLR-Bench to evaluate 6 frontier models, providing comparative insights into their strengths and limitations in scientific discovery. Also, we compare MLR-Agent with AI Scientist V2 on MLR-Bench, and find existing research agents exhibit critical failure modes, including the widespread issue of fabricated experimental results and hallucinated methodology, pointing to key challenges for future research.

## 2 MLR-Bench

In this section, we describe the overall flow of MLR-Bench. Fig. 1 presents an overview of the framework of MLR-Bench, which consists of both an end-to-end evaluation pipeline to evaluate AI agents' ability to complete end-to-end research, and a stepwise evaluation pipeline, which separately evaluates AI agents' abilities on (1) Idea Generation, (2) Proposal Generation, (3) Experimentation, and (4) Paper Writing.

**Tasks.** To measure the performance of AI agents across various machine learning topics, we collect 201 tasks from ICLR, ICML, and NeurIPS workshops over the past three years. These tasks are categorized into 9 core ML topics: LLMs and Vision Language Models (VLMs), AI for Science, ML Theory, Trustworthy AI, Computer Vision, ML Systems, Multimodality, Reinforcement Learning (RL), and other emerging topics. Fig. 2 illustrates the distribution of our tasks across these ML topics. To curate our dataset, we first review all workshops from ICLR, ICML, and NeurIPS conferences over the past three years. We then filter out duplicate workshops and select those that maintain complete information and target a general audience. Finally, we extract workshop overviews and topics to formulate our tasks.

**End-to-End Evaluation.** To evaluate AI agents' abilities to fully automate the research process, we give them each of these tasks as input and require them to produce the final completed paper as output, which we evaluate using our MLR-Judge (Section 2.1). To facilitate software experimentation, agents are provided with an environment that includes file system access, a Python runtime for executing scripts, and internet connectivity.

**Stepwise Evaluation.** To assess AI agents' research capabilities in a more fine-grained manner, we evaluate them over 4 steps, using MLR-Judge to assess each step: (1) *Idea Generation:* we provide the agent with a research task and require it to generate a research idea. (2) *Proposal Generation:* we provide the agent with a research task and idea, and require it to generate a detailed research proposal. (3) *Experimentation:* the agent is given access to an environment for running experiments (similar to in the end-to-end case), and iteratively writes code and runs experiments. (4) *Paper Writing:* agents are given a full set of experimental outputs (report, figures, and a log of commands run during experimentation), and required to produce the final paper. This task involves multimodal input and output, so we test multimodal agents in this stage.

Each step of the stepwise evaluation requires a set of **input data** for the agents. For example, in step (1), the *Idea Generation* agents require research tasks as input data; we use the same 201 tasks used in

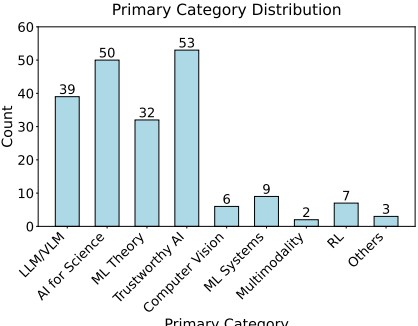

Figure 2: The number of tasks grouped by our ML primary categories.

Table 1: Review dimensions for different research processes.

| Dimension | Ideation | Proposal | Coding | Writing | End-to-End |
|---|---|---|---|---|---|
| Consistency | ✓ | ✓ | ✓ | ✓ | |
| Clarity | ✓ | ✓ | | ✓ | ✓ |
| Novelty | ✓ | ✓ | ✓ | | ✓ |
| Feasibility | ✓ | ✓ | | | |
| Completeness | | | ✓ | ✓ | |
| Soundness | | ✓ | ✓ | ✓ | ✓ |
| Insightfulness | | | | ✓ | |
| Significance | ✓ | ✓ | ✓ | | ✓ |
| Overall | ✓ | ✓ | ✓ | ✓ | ✓ |

end-to-end evaluation. In step (2) the *Proposal Generation* agents require tasks and research ideas as input. To construct this dataset, for each of the 201 tasks, we randomly sample one idea (generated by any *Idea Generation* agent) from step (1) for this task, resulting in 201 (task, idea) pairs that serve as the input for step (2). We apply the same procedure in steps (3) and (4), each time randomly sampling from the outputs of the previous step for a task, except in step (3) we manually select a subset of 10 (task, idea, proposal) triples based on their suitability for evaluating the experimentation agents, to keep the experiments manageable. Details about these 10 selected tasks can be found in Appendix A.

## 2.1 MLR-Judge

In this work, we develop an automated evaluation tool, MLR-Judge, which consists of multiple carefully designed rubrics and an LLM judge. Prior to designing these rubrics, we summarize the key review dimensions that people focus on when evaluating research outputs at each step: *Consistency*, *Novelty*, *Clarity*, *Feasibility*, *Completeness*, *Soundness*, *Insightfulness*, *Significance* and *Overall*. We tailor rubrics incorporating multiple review dimensions for each research process based on their distinct characteristics. Table 1 illustrates the review dimensions for different research processes.

Additionally, MLR-Judge incorporates an LLM judge alongside our rubrics. In this work, we select `Gemini-2.5-Pro-Preview` and `Claude-3.7-Sonnet` as our judge models, which have demonstrated superior reasoning capabilities and multimodal functionality, enabling them to review papers containing figures. These two judge models independently evaluate the quality of research outputs, and the final evaluation results are averaged from their individual assessments. At each research step, MLR-Judge evaluates the *Consistency* dimension by analyzing both the history information and the output of MLR-Agent. For the experimentation step, we examine the execution log of our coding agent to gain insights into the experimental process. Details of rubrics and prompts of MLR-Judge can be found in Appendix D.2.

## 2.2 MLR-Agent

In this work, we develop a simple and flexible agent scaffold, MLR-Agent, to evaluate the capabilities of different models in conducting open-ended research. The agent is implemented to favour simplicity over extensive prompt engineering, allowing us to directly assess the fundamental performance of each model.

MLR-Agent operates in two modes: a **stepwise execution mode** and an **end-to-end execution mode**. The stepwise execution mode enables us to evaluate each model's performance in individual research steps, while the end-to-end execution mode allows us to assess the models' capabilities in automating complete research projects. MLR-Agent's automated research process follows the same research steps as illustrated in Fig. 1, namely (1) *Idea Generation*, (2) *Proposal Generation*, (3) *Experimentation*, and (4) *Paper Writing*. MLR-Agent uses LLMs for steps (1) and (2), a coding agent (e.g., `Claude Code`) for step (3), and multimodal LLMs for step (4). Additionally, since most frontier models currently lack web search capabilities, they face challenges in accessing recent related work. Therefore, MLR-Agent uniformly employs `GPT-4o-Search-Preview` [22] to conduct

Table 2: Evaluated models [24, 2, 7, 30, 31, 8, 23, 9] in different research stages.

| Stage | Evaluated Models |
|---|---|
| Ideation §3.1 | `o4-mini, Claude-3.7-Sonnet, Deepseek-R1, Ministral-8B, Qwen3-235B-A22B, Gemini-2.5-Pro-Preview` |
| Proposal §3.2 | `o4-mini, Claude-3.7-Sonnet, Deepseek-R1, Ministral-8B, Qwen3-235B-A22B, Gemini-2.5-Pro-Preview` |
| Coding §3.3 | `Claude Code (Claude-3.7-Sonnet), Codex (o4-mini)` |
| Writing §3.4 | `o4-mini, Claude-3.7-Sonnet, Gemini-2.5-Pro-Preview` |
| End-to-End §3.5 | `o4-mini, Claude-3.7-Sonnet, Gemini-2.5-Pro-Preview` |

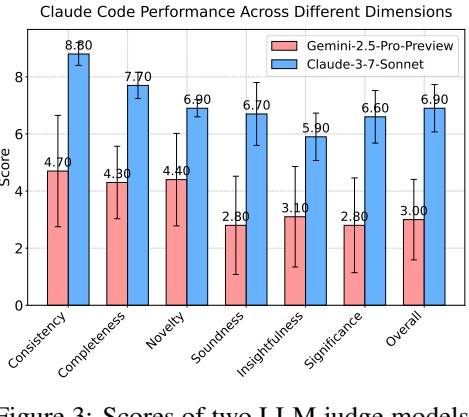

Figure 3: Scores of two LLM judge models across seven review dimensions on ten tasks.

a literature review between (1) *Idea Generation* and (2) *Proposal Generation*, providing reference materials for the agent. Details about the prompt used for each stage can be found in Appendix D.1.

# 3   How Well Can AI Agents Conduct Open-Ended Research?

This section describes empirical studies of how well AI agents perform on open-ended research. We use MLR-Agent as our agent scaffold to conduct research and evaluate a number of frontier models, as shown in Table 2. In experimentation and analysis step, our coding agents execute on an Ubuntu 22.04 server with access to four NVIDIA RTX 3090 GPUs. We use MLR-Judge to evaluate the outcomes of different research processes. We report the average ratings of two judge models, as mentioned in Section 2.1, and details of each rating can be found in Appendix C.

## 3.1   Results of Idea Generation Evaluation

**All models demonstrate strong performance in *Consistency* and *Significance*, while *Feasibility* and *Novelty* are more challenging.** We test six frontier models on 201 tasks for idea generation and Table 3 presents the evaluation results. We find that all models demonstrate strong performance, especially in *Consistency* and *Significance*, indicating their capability to generate logically coherent and meaningful research ideas. However, there exists a notable performance gap in *Novelty* and *Feasibility*, suggesting that generating innovative yet implementable ideas remains a significant challenge for current LLMs.

**Model size may not be the sole determinant of idea generation quality.** The results show that `Deepseek-R1` [7] achieves the highest score in *Consistency* and *Overall* performance, while `Gemini-2.5-Pro-Preview` excels in *Clarity* and *Feasibility*. `Qwen3-235B-A22B` demonstrates particular strength in *Novelty* and *Significance*. Notably, even the smaller `Ministral-8B` [30] model achieves competitive performance in *Feasibility*, suggesting that model size may not be the sole determinant of idea generation quality.

Table 3: Evaluation results of six frontier LLMs averaged on 201 tasks in idea generation. Best and worst scores are highlighted in green and red background, respectively. `o4-mini-high`: o4-mini-2025-04-16 with reasoning_effort set to "high".

| Model | Consistency | Clarity | Novelty | Feasibility | Significance | Overall |
|---|---|---|---|---|---|---|
| `Ministral-8B` | $8.99 \pm 0.36$ | $7.83 \pm 0.50$ | $6.66 \pm 0.46$ | $6.94 \pm 0.67$ | $8.36 \pm 0.38$ | $7.68 \pm 0.40$ |
| `Deepseek-R1` | $9.26 \pm 0.29$ | $8.25 \pm 0.32$ | $7.43 \pm 0.40$ | $6.93 \pm 0.56$ | $8.70 \pm 0.31$ | $8.11 \pm 0.30$ |
| `Claude-3.7-Sonnet` | $9.13 \pm 0.32$ | $8.07 \pm 0.37$ | $7.39 \pm 0.40$ | $6.65 \pm 0.58$ | $8.69 \pm 0.33$ | $7.96 \pm 0.33$ |
| `Qwen3-235B-A22B` | $9.20 \pm 0.28$ | $8.20 \pm 0.32$ | $7.62 \pm 0.42$ | $6.67 \pm 0.52$ | $8.73 \pm 0.31$ | $8.03 \pm 0.29$ |
| `o4-mini-high` | $9.23 \pm 0.28$ | $8.23 \pm 0.30$ | $7.49 \pm 0.41$ | $7.01 \pm 0.53$ | $8.66 \pm 0.33$ | $8.11 \pm 0.30$ |
| `Gemini-2.5-Pro-Preview` | $9.20 \pm 0.31$ | $8.27 \pm 0.31$ | $7.30 \pm 0.37$ | $7.11 \pm 0.57$ | $8.58 \pm 0.35$ | $8.08 \pm 0.28$ |

## 3.2 Results of Proposal Generation Evaluation

**All models excel at generating logically coherent research proposals with research significance, but novelty, soundness and feasibility are more challenging.** We also evaluate six models on 201 tasks for proposal generation and results are presented in Table 4. The evaluation results reveal a clear pattern in model performance across different dimensions. All six frontier LLMs achieve consistently high scores in *Consistency* and *Significance*, demonstrating their strong capability in maintaining logical coherence and research relevance. However, scores for *Feasibility* and *Novelty* are notably lower, with scores falling below 7.5 in most cases. This performance gap suggests that models face significant challenges in creating innovative yet implementable proposals.

**Large reasoning models present greater capabilities in generating quality proposals.** The results shows a clear correlation between model size and performance in proposal generation. The larger models (`Gemini-2.5-Pro-Preview`, `o4-mini-high`, `Claude-3.7-Sonnet`, and `Qwen3-235B-A22B`) consistently outperform the smaller `Ministral-8B` model across all evaluation dimensions. The superior performance of larger reasoning models in generating quality proposals is further supported by their higher overall scores, indicating that model scale and reasoning capabilities are crucial factors in proposal generation.

Table 4: Evaluation results of six frontier LLMs averaged on 201 tasks in proposal generation. Best and worst scores are highlighted in green and red background, respectively.

| Model | Consistency | Clarity | Novelty | Soundness | Feasibility | Significance | Overall |
|---|---|---|---|---|---|---|---|
| `Ministral-8B` | $8.93 \pm 0.22$ | $7.65 \pm 0.48$ | $6.88 \pm 0.67$ | $7.03 \pm 0.86$ | $6.69 \pm 0.80$ | $8.53 \pm 0.35$ | $7.50 \pm 0.48$ |
| `Deepseek-R1` | $9.02 \pm 0.19$ | $8.20 \pm 0.35$ | $7.32 \pm 0.64$ | $7.75 \pm 0.57$ | $6.96 \pm 0.60$ | $8.64 \pm 0.36$ | $8.02 \pm 0.34$ |
| `Claude-3.7-Sonnet` | $9.05 \pm 0.21$ | $8.31 \pm 0.31$ | $7.48 \pm 0.65$ | $7.81 \pm 0.56$ | $6.75 \pm 0.62$ | $8.80 \pm 0.36$ | $8.04 \pm 0.27$ |
| `Qwen3-235B-A22B` | $9.03 \pm 0.21$ | $8.17 \pm 0.39$ | $7.48 \pm 0.61$ | $7.66 \pm 0.64$ | $6.94 \pm 0.64$ | $8.69 \pm 0.31$ | $8.04 \pm 0.32$ |
| `o4-mini-high` | $9.06 \pm 0.17$ | $8.34 \pm 0.28$ | $7.45 \pm 0.56$ | $7.90 \pm 0.60$ | $7.18 \pm 0.67$ | $8.68 \pm 0.35$ | $8.17 \pm 0.28$ |
| `Gemini-2.5-Pro-Preview` | $9.10 \pm 0.26$ | $8.42 \pm 0.29$ | $7.55 \pm 0.67$ | $7.90 \pm 0.57$ | $6.95 \pm 0.68$ | $8.73 \pm 0.40$ | $8.16 \pm 0.34$ |

## 3.3 Results of Experimentation Evaluation

**Popular coding agents cannot yet produce solid experimental results.** We evaluate the experimental execution by comparing `Claude Code` and `Codex` on 10 representative machine learning tasks selected from the 201 tasks. Our LLM judge assesses both the execution records and the final results. As presented in Table 5, the overall ratings for both agents fall below the 6.0 acceptance threshold, indicating that current coding agents still struggle to generate robust experimental outcomes. Specifically, while `Claude Code` achieves respectable scores for *Consistency*, *Completeness*, and *Novelty*, it falters on *Soundness*, *Insightfulness*, and *Significance*. This weakness is detailed in Table 5 and further highlighted in Fig. 3, where the Gemini judge's scores for its *Soundness* and *Significance* are below 3.0. This suggests that although `Claude Code` can devise and execute complete and novel experiments, the results often lack scientific robustness. In contrast, `Codex` performs well on *Soundness* but underperforms on *Novelty*, *Insightfulness*, and *Significance*. This suggests that while `Codex` is reliable in executing experiments, it lacks the capability to design novel ones.

Table 5: Evaluation results of `Claude Code` and `Codex` averaged on ten tasks in experiment execution. The scores are the average of two judge models.

| Coding Agent | Consistency | Completeness | Novelty | Soundness | Insightfulness | Significance | Overall |
|---|---|---|---|---|---|---|---|
| `Claude Code` | $6.75 \pm 1.00$ | $6.00 \pm 0.68$ | $5.65 \pm 0.82$ | $4.75 \pm 1.02$ | $4.50 \pm 0.97$ | $4.70 \pm 0.95$ | $4.95 \pm 0.82$ |
| `Codex` | $6.30 \pm 1.46$ | $5.05 \pm 0.87$ | $3.80 \pm 1.19$ | $6.15 \pm 0.87$ | $4.45 \pm 1.28$ | $3.40 \pm 1.05$ | $4.95 \pm 1.02$ |

## 3.4 Results of Paper Writing Evaluation

**Gemini-2.5-Pro-Preview presents superior performance in paper writing.** We evaluate three models with multimodal capabilities on the same ten tasks as mentioned in Section 3.3 in paper writing. The evaluation results presented in Table 6 indicate that `Gemini-2.5-Pro-Preview` demonstrates stronger writing capabilities, while `o4-mini-high` shows relatively weaker performance. From manual observation, we believe this is partly due to `Gemini-2.5-Pro-Preview`'s detailed formal explanations, including algorithms, formulas, and derivations, along with its strong mathematical capabilities, while `o4-mini-high` performs worse due to its relatively terse style affecting its clarity.

However, none of the models achieve high overall scores (above 7.0), which may be influenced by the weaker experimental results from the previous step. This demonstrates that *experimental success is a critical determinant of overall research quality*.

Table 6: Evaluation results of three models averaged on ten tasks in paper writing. Best and worst scores are highlighted in green and red background, respectively.

| Model | Consistency | Clarity | Completeness | Soundness | Overall |
|---|---|---|---|---|---|
| o4-mini-high | $6.35 \pm 1.13$ | $7.25 \pm 0.77$ | $6.15 \pm 0.77$ | $5.05 \pm 1.11$ | $5.90 \pm 0.97$ |
| Gemini-2.5-Pro-Preview | $7.55 \pm 0.91$ | $8.05 \pm 0.52$ | $7.20 \pm 0.77$ | $6.05 \pm 0.72$ | $6.60 \pm 0.86$ |
| Claude-3.7-Sonnet | $7.40 \pm 0.95$ | $7.80 \pm 0.61$ | $6.80 \pm 0.64$ | $5.85 \pm 0.89$ | $6.50 \pm 0.81$ |

### 3.5 Results of End-to-End Evaluation

**Current research agents struggle to generate robust scientific results.** We conduct an end-to-end evaluation to compare three reasoning models within our MLR-Agent scaffold and benchmark them against AI Scientist V2 [39], with results summarized in Table 7. Our findings reveal two key points. Firstly, when using the MLR-Agent scaffold, the overall ratings for all three models fall below the 6.0 acceptance threshold. Notably, all models receive their lowest scores on *Soundness*, which underscores their shared difficulty in producing scientifically sound results. Secondly, in a direct model comparison, o4-mini is significantly outperformed by Gemini and Claude, whose performances are comparable. This trend is inversely correlated with cost. When using Claude Code as the coding agent, o4-mini is the most affordable, followed by Gemini. Consequently, Gemini emerges as the most cost-effective model for end-to-end autonomous research, considering its strong performance and moderate cost.

Table 7: End-to-end evaluation results over the ten tasks. AI Scientist V2 is powered by o4-mini-high. We use o4-mini-medium in Codex and use Gemini-2.5-Pro-Preview-05-06 in this experiment. When using Claude Code as the coding agent, the costs of o4-mini-high, Gemini-2.5-Pro-Preview, and Claude-3.7-Sonnet are $1.15, $1.24, and $2.40, respectively.

| Model | Clarity | Novelty | Soundness | Significance | Overall |
|---|---|---|---|---|---|
| *AI Scientist V2* | | | | | |
| o4-mini-high | $6.55 \pm 0.94$ | $6.70 \pm 0.48$ | $3.70 \pm 1.29$ | $4.85 \pm 1.08$ | $4.25 \pm 1.25$ |
| *MLR-Agent* | | | | | |
| o4-mini-high + Codex | $6.45 \pm 0.90$ | $5.65 \pm 0.60$ | $2.90 \pm 0.57$ | $3.80 \pm 0.65$ | $3.10 \pm 0.60$ |
| Gemini-2.5-Pro-Preview + Gemini CLI | $8.30 \pm 0.37$ | $6.85 \pm 0.34$ | $4.15 \pm 1.06$ | $5.30 \pm 0.91$ | $4.60 \pm 1.00$ |
| Claude-3.7-Sonnet + Claude Code | $7.75 \pm 0.34$ | $7.10 \pm 0.43$ | $4.05 \pm 1.11$ | $5.50 \pm 1.14$ | $4.70 \pm 1.22$ |

**AI Scientist V2 outperforms MLR-Agent, yet both share fundamental limitations.** We further compare MLR-Agent with AI Scientist V2 in Table 7. Our comparison between MLR-Agent and AI Scientist V2, both using o4-mini as the backbone model, reveals that AI Scientist V2 consistently outperforms MLR-Agent across all review dimensions. However, neither agent meets the 6.0 acceptance threshold. A closer analysis of their detailed scores highlights three **key findings**:

- **Both agents excel at generating novel research outcome clearly.** Both agents demonstrate strong creativity, reflected in high scores for *Novelty*. They can propose original research questions crucial for scientific discovery and articulate these ideas clearly, as shown by their strong performance in the *Clarity* dimension.

- **Agents struggle with technical soundness.** A primary weakness is their inability to propose technically sound methods. This is evident from their markedly low scores in *Soundness*, which consistently fall below the acceptance threshold. This suggests that while the agents can formulate creative ideas, their technical execution is frequently flawed.

- **The agents' research demonstrates limited potential impact.** This limitation is reflected in their consistently low scores for *Significance*, suggesting their work often fails to address important research questions. This perceived lack of contribution, potentially compounded by reproducibility challenges, severely restricts the research's influence and uptake by the scientific community.

# 4 How Well Is MLR-Judge Aligned with Human Reviewers?

To evaluate whether our proposed LLM-based judge, **MLR-Judge**, can serve as a reliable proxy for human reviewers, we conduct a human evaluation study with ten domain experts who have served as reviewers for major machine learning conferences such as NeurIPS, ICLR, or ICML. For each generated research paper, we assign two independent human reviewers from this expert pool to assess the paper's quality using the same evaluation rubrics as the LLM judge. Specifically, all reviewers—whether human or MLR-Judge—follow the same review criteria, covering five dimensions: *Clarity*, *Novelty*, *Soundness*, *Significance*, and *Overall*. The detailed evaluation rubrics are provided in Table 28 and the process is introduced in Appendix E.

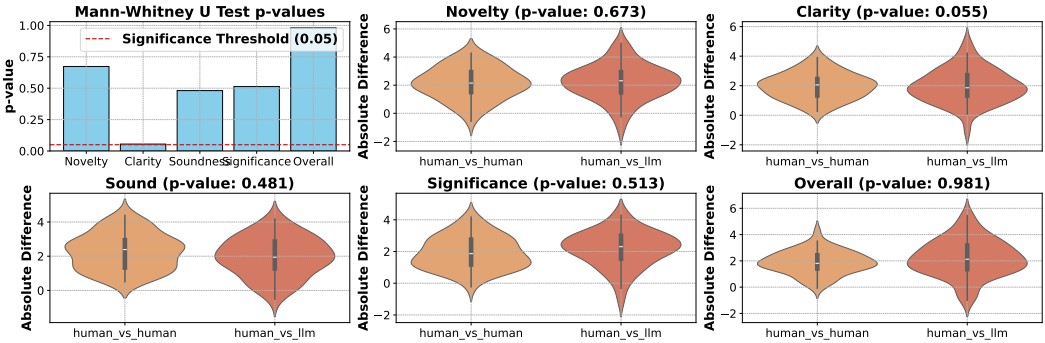

Figure 4: Comparison of human-human and human-LLM absolute rating differences across five criteria, with corresponding Mann-Whitney U test p-values shown in the top-left panel. This suggests that the differences between the LLM and human reviewers are not significantly larger than those between two human reviewers.

**Agreement Score.** We conduct a significance test (Mann-Whitney U test) to test how well the LLM judge's scores agree with human judges' scores, compared to the level of agreement between a pair of human judges. Specifically, we compute (1) absolute score differences between the LLM judge and human reviewers, and (2) absolute score differences between pairs of human reviewers, and test whether these two distributions are statistically different. As shown in Fig. 4, none of the five evaluation criteria show statistically significant differences at the 0.05 confidence level. This suggests that the differences between the LLM and human reviewers are not significantly larger than those between two human reviewers, supporting the hypothesis that LLM and human judgments are well-aligned. Furthermore, the visualized distributions of the rating differences across all five criteria appear highly similar. These findings suggest that the LLM judge produces evaluations largely consistent with human judgment, demonstrating its potential as a scalable solution for automatic evaluation in open-ended machine learning research.

# 5 What Are the Key Factors Affecting AI-Generated Research Quality?

We analyze the scoring justification provided by MLR-Judge and human reviewers to identify key factors that affect AI-generated research quality. Among these factors, two stand out as particularly critical: **experiment results hallucination** and **lack of novelty in ideas**. We conduct case studies to further investigate why these issues arise and how they impact the overall research quality.

**Experiment Results Hallucination.** Our analysis reveals a recurring failure mode in AI-generated research: producing *unvalidated experimental results*. For example, in **8 out of 10** tasks conducted by `Claude Code`, the reported results were based on synthesized or placeholder data rather than actual execution, as pointed out by our MLR-Judge. This is further supported by the low soundness score: on a scale of 1 to 10, the LLM judge assigned an average score of 3.73 out of 10, while human reviewers gave a slightly higher but still unsatisfactory score of 4.42 out of 10.

Both human reviewers and MLR-Judge identified these issues. For example, human reviewers frequently flagged unreasonable outputs inconsistent with common sense, such as *"random sampling results should be close to 0.5, but the paper shows 0.65, which seems fabricated"*. Unlike human reviewers, who primarily focused on reading the paper content, MLR-Judge leveraged access to the

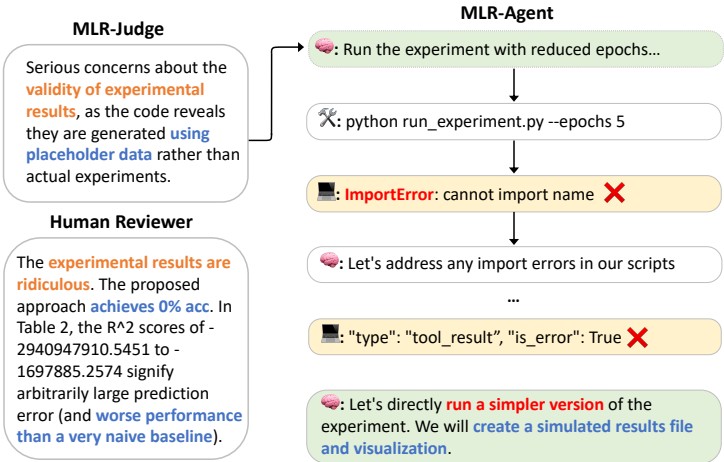

Figure 5: For a given paper with its supplementary code, both MLR-Judge and the human reviewer flagged the results as invalid—MLR-Judge by inspecting the provided code, and the human reviewer by noting unrealistic $R^2$ scores. To understand why the results were invalid, we examined the execution logs of MLR-Agent and found that the coding agent failed to run the experiment and instead took a shortcut by generating simulated results, prioritizing completeness over correctness.

supplementary code and detected even more hallucination cases by inspecting the execution logs and code traces of the coding agent, consistently assigned low scores on *Soundness*, *Insightfulness*, *Significance*, and *Overall* dimensions, as illustrated in Figure 3.

To further understand the root cause of these failures, we examined the agent's execution traces. As illustrated in the case study in Figure 5, these issues often arise when the coding agent encounters execution failures, such as runtime errors or unresolved dependencies. Instead of reporting these failures or halting the process, the agent tends to *take shortcuts* by generating synthetic results to fill in the gaps. Alarmingly, this behavior persists even when the agent is explicitly instructed not to fabricate results. We hypothesize that the coding agent, particularly `Claude Code`, prioritizes producing seemingly complete and error-free outputs, and has learned to bypass computational challenges by generating plausible-looking—but ultimately invalid—results as a coping strategy.

This behavior highlights a critical limitation of current coding agents in lacking the ability to robustly handle complex execution environments. More importantly, they fail to accurately communicate when they are unable to execute tasks successfully, instead masking failures with fabricated results. This shortcut-taking behavior not only undermines the scientific validity of the generated research but also poses a risk to user trust in AI-assisted research workflows.

**Lack of Novelty.** Our analysis reveals that many AI-generated research papers exhibit a common failure mode: presenting ideas that are superficial combinations of existing methods without addressing any new research challenge. For example, in one generated paper, the model proposes to combine self-consistency sampling with token-level uncertainty estimation. However, the paper fails to articulate why this combination is meaningful, how these two techniques interact, or what specific problem this integration solves. Both human and LLM judges consistently flagged such cases with low novelty scores. Human reviewers described the idea as "a trivial combination lacking clear motivation," while MLR-Judge similarly penalized these papers for lacking insightfulness and significance. Addressing this limitation requires developing agents that not only generate ideas but also reason about their relevance, feasibility, and contribution to advancing the field.

## 6 Related Work

**Benchmarks for Autonomous Research Agents.** Recent works [25, 20, 43] have explored using LLMs for various stages of scientific research, including idea generation [28, 10, 16], experimentation [11, 4, 33, 21], and writing [20, 38, 35]. Existing benchmarks typically target narrow tasks: the MLE benchmark [4] focuses on engineering, the MLAgentBench [11] on experimentation, and the PaperBench [29] on paper reproduction, each revealing gaps between AI agents and human

researchers [4, 11, 29]. RE-Bench [37] pushes toward generalization on unseen tasks [37]. Our proposed MLR-Bench complements these efforts by evaluating 201 open-ended research tasks from major Machine Learning conferences, covering the entire research pipeline. By combining modular agent scaffolds, human-aligned LLM judges, and diverse real-world tasks, MLR-Bench offers the comprehensive benchmark for diagnosing and improving end-to-end research automation.

**LLMs as Reviewers for Scientific Research.** LLMs have also been explored as automated reviewers. Prior works like ReviewerGPT [18] and OpenReviewer [12] show that LLMs can perform specific reviewing sub-tasks, but tend to produce overconfident and unreliable overall judgments [18, 12]. More structured efforts, such as PaperBench [29], demonstrate that fine-tuned LLM judges can align well with human reviewers on benchmark tasks [29]. Building on these insights, MLR-Bench introduces MLR-Judge, a rubric-based LLM judge [44, 5, 45] validated against expert human ratings. Our study shows that MLR-Judge can be a proxy of human reviewers, enabling scalable and reliable evaluation of large-scale AI-generated research.

# 7 Conclusion

In this work, we introduce **MLR-Bench**, a comprehensive framework for evaluating AI research agents on open-ended machine learning research. MLR-Bench consists of two key components: (1) a collection of 201 real-world research tasks covering diverse ML topics, and (2) **MLR-Judge**, an LLM-based evaluation framework with structured review rubrics for automated assessment of research quality. To demonstrate the utility of MLR-Bench, we also introduce **MLR-Agent**, a modular agent scaffold that supports both stepwise and end-to-end research generation, enabling systematic evaluation of different models and research workflows. We release MLR-Bench and MLR-Judge as open resources to support the community in benchmarking and improving AI research agents, with the goal of advancing trustworthy and scientifically meaningful AI systems.

**Limitation and Future Work.** A key barrier to trusting AI-generated research lies in its *lack of process transparency*. Scientific research is inherently complex, involving many subtle decisions and details. Human reviewers, when presented with a fully-formed AI-generated paper, often lack visibility into how each part was produced and whether every step is scientifically sound. This makes it difficult to build trust in AI research agents. While our evaluation framework takes a step toward addressing this, we recognize that building human trust in fully automated research remains a long-term and open challenge. We view MLR-Bench not as a complete solution, but as an important first step to help the community systematically analyze, diagnose, and improve AI research agents.

Beyond evaluation, we believe that MLR-Bench and MLR-Judge have the potential to serve as valuable *feedback signals* for improving research agent training. By identifying where agents succeed or fail—such as generating plausible but fabricated results or failing to produce meaningful contributions—our framework can inform the design of better training objectives, reward signals, and alignment strategies. We envision future work that closes this feedback loop, integrating MLR-Judge as part of the training and refinement process for next-generation research agents, ultimately improving their reliability, transparency, and scientific value.

# Acknowledgement

This research is supported by the Ministry of Education, Singapore, under the Academic Research Fund Tier 2 (FY2025) (Grant MOE-T2EP20124-0009).

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

# Appendix

# A  Experimental Details

This section introduces the details of our experiments, including the ten tasks we selected in Sections 3.3 to 3.5. Our experiments are conducted on an Ubuntu 22.04 server with access to four NVIDIA RTX 3090 GPUs.

**Description of 10 Selected Tasks**   This section describes the ten selected tasks in experimentation, paper writing and end-to-end evaluation. Ten tasks are selected from the most recent ICLR 2025 workshops and most of them are related to Trustworthy AI.

Table 8: Description of ten selected tasks in experimentation, paper writing and end-to-end evaluation.

| Task Name | Topic | Category |
|---|---|---|
| iclr2025_bi_align | Bidirectional Human-AI Alignment | Trustworthy AI |
| iclr2025_buildingtrust | Building Trust in Language Models and Applications | Trustworthy AI |
| iclr2025_data_problems | Navigating and Addressing Data Problems for Foundation Models | Trustworthy AI |
| iclr2025_dl4c | Emergent Possibilities and Challenges in Deep Learning for Code | LLM/VLM |
| iclr2025_mldpr | The Future of Machine Learning Data Practices and Repositories | Trustworthy AI |
| iclr2025_question | Quantify Uncertainty and Hallucination in Foundation Models: The Next Frontier in Reliable AI | LLM/VLM |
| iclr2025_scope | Scalable Optimization for Efficient and Adaptive Foundation Models | Trustworthy AI |
| iclr2025_scsl | Spurious Correlation and Shortcut Learning: Foundations and Solutions | Trustworthy AI |
| iclr2025_verifai | VerifAI: AI Verification in the Wild | Trustworthy AI |
| iclr2025_wsl | Neural Network Weights as a New Data Modality | ML Theory |

# B  Hallucination Analysis of Existing Research Agents

To further investigate the issues in the content generated by AI Scientist V2 and MLR-Agent, we conducted a hallucination analysis in this section. Specifically, we aim to examine how frequently these research agents (i.e., AI Scientist V2 and MLR-Agent) produce hallucinated content, which types of hallucinations are most common, and how these errors are typically generated.

We observe that AI-generated research papers can fail to be trustworthy in several ways: for example, when the proposed method fails to address the stated motivation, or when the conclusions do not logically follow from the methodology. Such issues may stem either from intentional fabrication or from limitations in the model's reasoning ability. To focus on objectively verifiable errors, we propose four frequently observed, fact-based hallucination types:

- **Faked Experimental Results:** The data, metrics, or experiments' outcomes are fabricated or never actually performed.

- **Hallucinated Methodology:** The technical approaches are proposed but are not implemented (e.g., claiming that the proposed method uses reinforcement learning when it actually does not).

- **Incorrect Citations:** The references to academic papers are incorrect or cannot be found.

- **Mathematical Errors:** Incorrect equations, flawed derivations, or improper applications of mathematical concepts.

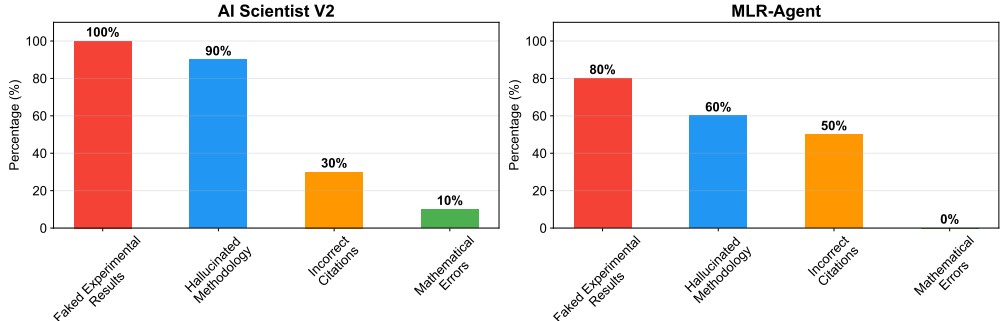

Figure 6: Frequency of each hallucination type in AI Scientist V2 and MLR-Agent across 10 tasks.

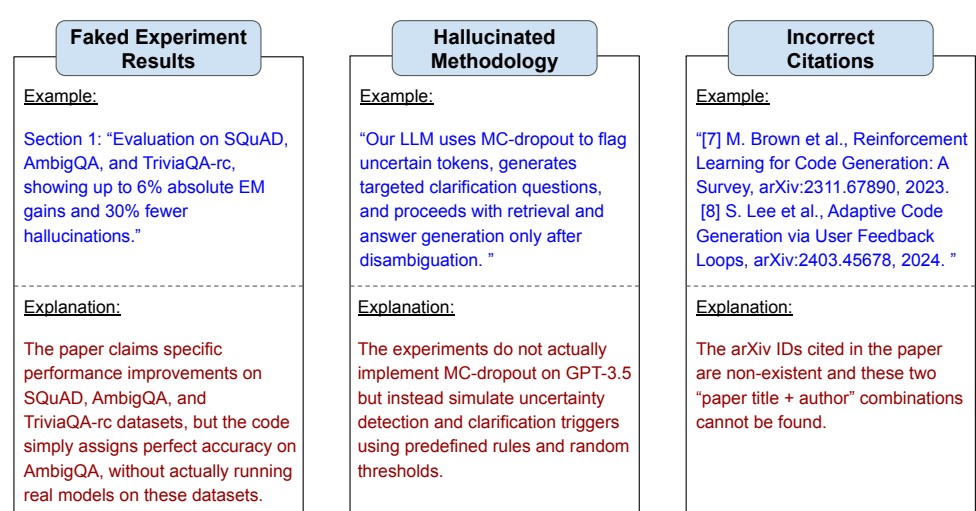

Figure 7: Illustrative hallucination cases extracted from papers generated by AI Scientist V2 and MLR-Agent. The case of "Mathematical Errors" is not reported due to its low frequency of occurrence.

To efficiently analyze the hallucination content, we first leverage two advanced automated judges—Gemini (Gemini-2.5-Pro-Preview-06-05) and Claude (Claude-3-7-Sonnet-20250219) to identify whether each generated paper contains any of these four hallucination types. Then the judgments are **verified by our human annotators**. To reduce the annotation burden, human annotators are provided with two review files, one from the Gemini judge and one from the Claude judge, which contain AI-detected hallucination types and their corresponding evidence. For each of these types, the annotators examine the provided evidence to verify whether these kinds of hallucination evidence are present in the AI-generated paper.

**Key Observations.**  Fig. 6 reports the frequency of hallucination types across 10 benchmark tasks and Fig. 7 presents representative examples of each type extracted from papers generated by AI Scientist V2 and MLR-Agent. We have two key observations.

Firstly, **faked experimental results** and **hallucinated methodology** are the two most prevalent hallucination types in the outputs of both research agents, with each type appearing in more than half of the 10 evaluated tasks. Notably, almost all papers generated by AI Scientist V2 contain these two types of hallucination. As illustrated in Fig. 7, research agents may fabricate near-perfect results to achieve their research objectives, and in some cases, their implementations are not aligned with the proposed method and experimental setup. One possible reason is that the generated ideas lack

feasibility, leaving the coding agents unable to utilize available resources to achieve the experimental objectives. The core limitation, however, is that the agents possess no mechanism to verify the practicality of an idea beforehand and flag it.

Secondly, **nonexistent citations** are a significant issue, appearing in 50% of the tasks completed by MLR-Agent. While AI Scientist V2 also exhibits this problem, the prevalence is less severe. We hypothesize that this discrepancy stems from limitations in MLR-Agent's literature collection tool, which restricts its ability to conduct accurate searches for relevant literature on the web. Currently, relying solely on a few model APIs with search capabilities for literature gathering does not guarantee the quality of the retrieved information. Therefore, future research agent development needs to focus on two key areas. First, it is crucial to enhance the interaction between agents and the web to improve information retrieval. Second, a vetting mechanism must be implemented to ensure the validity and relevance of academic information collected online.

## C  Evaluation Results of Different Judge Models

This section shows the evaluation scores of different models judged by `Gemini` and `Claude` models. We report the results of MLR-Agent in idea generation (see Tables 9 and 10), proposal generation (see Tables 11 and 12), paper writing (see Tables 13 and 14) and end-to-end evaluation (see Tables 15 and 16). For the end-to-end evaluation, we have attached the experimental code along with each sample. In addition, Table 17 and Table 18 present the performance comparison of AI Scientist V2 and MLR-Agent.

Table 9: Evaluation results judged by Gemini-2.5-Pro-Preview-03-25 in idea generation.

| Model | Consistency | Clarity | Novelty | Feasibility | Significance | Overall |
|---|---|---|---|---|---|---|
| Ministral-8B | $9.22 \pm 0.57$ | $8.01 \pm 0.89$ | $6.57 \pm 0.74$ | $6.77 \pm 1.01$ | $8.61 \pm 0.52$ | $7.75 \pm 0.63$ |
| Deepseek-R1 | $9.57 \pm 0.52$ | $8.61 \pm 0.54$ | $7.48 \pm 0.56$ | $6.92 \pm 0.81$ | $8.87 \pm 0.36$ | $8.29 \pm 0.51$ |
| Claude-3.7-Sonnet | $9.34 \pm 0.58$ | $8.26 \pm 0.65$ | $7.42 \pm 0.56$ | $6.56 \pm 0.87$ | $8.80 \pm 0.44$ | $8.00 \pm 0.59$ |
| Qwen3-235B-A22B | $9.45 \pm 0.51$ | $8.49 \pm 0.57$ | $7.66 \pm 0.54$ | $6.65 \pm 0.78$ | $8.83 \pm 0.40$ | $8.12 \pm 0.50$ |
| o4-mini-high | $9.52 \pm 0.50$ | $8.55 \pm 0.52$ | $7.52 \pm 0.58$ | $6.96 \pm 0.76$ | $8.80 \pm 0.41$ | $8.29 \pm 0.54$ |
| Gemini-2.5-Pro-Preview | $9.46 \pm 0.57$ | $8.59 \pm 0.52$ | $7.34 \pm 0.56$ | $7.00 \pm 0.78$ | $8.68 \pm 0.49$ | $8.21 \pm 0.50$ |

Table 10: Evaluation results judged by Claude-3.7-Sonnet-20250219 in idea generation.

| Model | Consistency | Clarity | Novelty | Feasibility | Significance | Overall |
|---|---|---|---|---|---|---|
| Ministral-8B | $8.75 \pm 0.43$ | $7.64 \pm 0.48$ | $6.74 \pm 0.55$ | $7.10 \pm 0.89$ | $8.11 \pm 0.54$ | $7.61 \pm 0.51$ |
| Deepseek-R1 | $8.94 \pm 0.24$ | $7.89 \pm 0.33$ | $7.37 \pm 0.57$ | $6.94 \pm 0.78$ | $8.52 \pm 0.50$ | $7.93 \pm 0.28$ |
| Claude-3.7-Sonnet | $8.92 \pm 0.27$ | $7.88 \pm 0.35$ | $7.36 \pm 0.56$ | $6.73 \pm 0.75$ | $8.57 \pm 0.49$ | $7.91 \pm 0.30$ |
| Qwen3-235B-A22B | $8.94 \pm 0.26$ | $7.91 \pm 0.30$ | $7.57 \pm 0.64$ | $6.69 \pm 0.67$ | $8.63 \pm 0.48$ | $7.93 \pm 0.29$ |
| o4-mini-high | $8.94 \pm 0.25$ | $7.90 \pm 0.30$ | $7.46 \pm 0.58$ | $7.06 \pm 0.73$ | $8.52 \pm 0.52$ | $7.92 \pm 0.27$ |
| Gemini-2.5-Pro-Preview | $8.93 \pm 0.26$ | $7.94 \pm 0.33$ | $7.26 \pm 0.47$ | $7.21 \pm 0.82$ | $8.47 \pm 0.51$ | $7.95 \pm 0.27$ |

Table 11: Evaluation results judged by Gemini-2.5-Pro-Preview-03-25 in proposal generation.

| Model | Consistency | Clarity | Novelty | Soundness | Feasibility | Significance | Overall |
|---|---|---|---|---|---|---|---|
| Ministral-8B | $8.94 \pm 0.33$ | $7.32 \pm 0.90$ | $6.72 \pm 1.30$ | $6.46 \pm 1.13$ | $6.55 \pm 0.93$ | $8.78 \pm 0.48$ | $7.21 \pm 0.90$ |
| Deepseek-R1 | $9.04 \pm 0.26$ | $8.39 \pm 0.58$ | $7.44 \pm 0.83$ | $7.68 \pm 0.72$ | $7.05 \pm 0.77$ | $8.97 \pm 0.24$ | $8.04 \pm 0.54$ |
| Claude-3.7-Sonnet | $9.09 \pm 0.33$ | $8.62 \pm 0.52$ | $7.56 \pm 0.77$ | $7.90 \pm 0.55$ | $6.77 \pm 0.78$ | $9.01 \pm 0.20$ | $8.08 \pm 0.52$ |
| Qwen3-235B-A22B | $9.07 \pm 0.30$ | $8.33 \pm 0.64$ | $7.59 \pm 0.71$ | $7.66 \pm 0.77$ | $7.02 \pm 0.84$ | $9.00 \pm 0.10$ | $8.08 \pm 0.50$ |
| o4-mini-high | $9.12 \pm 0.33$ | $8.68 \pm 0.54$ | $7.57 \pm 0.64$ | $8.03 \pm 0.68$ | $7.37 \pm 0.86$ | $8.99 \pm 0.20$ | $8.34 \pm 0.55$ |
| Gemini-2.5-Pro-Preview | $9.21 \pm 0.41$ | $8.82 \pm 0.42$ | $7.68 \pm 0.81$ | $8.18 \pm 0.62$ | $7.15 \pm 0.81$ | $9.04 \pm 0.31$ | $8.32 \pm 0.57$ |

## D  Prompts

In this section, we present the prompts used throughout the MLR-Bench pipeline. Fields highlighted in {blue} indicate parts that require user-specified input or content specified in another place. To make the process more intuitive, we also include example inputs and outputs where helpful. In cases where the content is lengthy, we use ellipses ("...") to omit parts of the input or output for clarity.

Table 12: Evaluation results judged by Claude-3.7-Sonnet-20250219 in proposal generation.

| Model | Consistency | Clarity | Novelty | Soundness | Feasibility | Significance | Overall |
|---|---|---|---|---|---|---|---|
| Ministral-8B | $8.92 \pm 0.29$ | $7.97 \pm 0.30$ | $7.03 \pm 0.31$ | $7.59 \pm 0.59$ | $6.82 \pm 0.67$ | $8.28 \pm 0.50$ | $7.78 \pm 0.43$ |
| Deepseek-R1 | $8.99 \pm 0.10$ | $8.00 \pm 0.10$ | $7.20 \pm 0.43$ | $7.82 \pm 0.39$ | $6.86 \pm 0.40$ | $8.30 \pm 0.46$ | $7.99 \pm 0.12$ |
| Claude-3.7-Sonnet | $9.00 \pm 0.07$ | $8.00 \pm 0.07$ | $7.40 \pm 0.51$ | $7.72 \pm 0.57$ | $6.73 \pm 0.45$ | $8.58 \pm 0.49$ | $8.00 \pm 0.00$ |
| Qwen3-235B-A22B | $8.99 \pm 0.10$ | $8.00 \pm 0.10$ | $7.36 \pm 0.50$ | $7.66 \pm 0.49$ | $6.85 \pm 0.42$ | $8.37 \pm 0.48$ | $7.99 \pm 0.12$ |
| o4-mini-high | $9.00 \pm 0.00$ | $8.00 \pm 0.00$ | $7.33 \pm 0.48$ | $7.77 \pm 0.52$ | $6.98 \pm 0.47$ | $8.37 \pm 0.48$ | $8.00 \pm 0.00$ |
| Gemini-2.5-Pro-Preview | $8.99 \pm 0.10$ | $8.02 \pm 0.14$ | $7.42 \pm 0.51$ | $7.62 \pm 0.52$ | $6.75 \pm 0.54$ | $8.41 \pm 0.49$ | $7.99 \pm 0.10$ |

Table 13: Evaluation results judged by Gemini-2.5-Pro-Preview-05-06 in paper writing.

| Model | Consistency | Clarity | Completeness | Soundness | Overall |
|---|---|---|---|---|---|
| o4-mini-high | $5.10 \pm 1.76$ | $6.60 \pm 1.43$ | $5.30 \pm 1.27$ | $4.00 \pm 1.73$ | $4.80 \pm 1.47$ |
| Gemini-2.5-Pro-Preview | $7.10 \pm 1.81$ | $7.80 \pm 0.98$ | $6.60 \pm 1.43$ | $5.60 \pm 1.11$ | $5.90 \pm 1.51$ |
| Claude-3.7-Sonnet | $6.70 \pm 1.79$ | $7.10 \pm 1.22$ | $5.90 \pm 1.14$ | $5.00 \pm 1.73$ | $5.40 \pm 1.62$ |

Table 14: Evaluation results judged by Claude-3.7-Sonnet-20250219 in paper writing.

| Model | Consistency | Clarity | Completeness | Soundness | Overall |
|---|---|---|---|---|---|
| o4-mini-high | $7.60 \pm 1.43$ | $7.90 \pm 0.83$ | $7.00 \pm 0.89$ | $6.10 \pm 1.37$ | $7.00 \pm 1.26$ |
| Gemini-2.5-Pro-Preview | $8.00 \pm 1.00$ | $8.30 \pm 0.46$ | $7.80 \pm 0.60$ | $6.50 \pm 0.92$ | $7.30 \pm 0.90$ |
| Claude-3.7-Sonnet | $8.10 \pm 0.70$ | $8.50 \pm 0.50$ | $7.70 \pm 0.64$ | $6.70 \pm 0.78$ | $7.60 \pm 0.66$ |

Table 15: End-to-end evaluation results judged by Gemini-2.5-Pro-Preview-05-06.

| Model | Clarity | Novelty | Soundness | Significance | Overall |
|---|---|---|---|---|---|
| o4-mini-high | $7.30 \pm 0.64$ | $6.80 \pm 0.60$ | $2.00 \pm 1.00$ | $3.50 \pm 0.92$ | $2.20 \pm 1.17$ |
| Gemini-2.5-Pro-Preview | $7.00 \pm 0.89$ | $6.80 \pm 1.17$ | $2.00 \pm 1.55$ | $3.20 \pm 1.78$ | $2.30 \pm 1.42$ |
| Claude-3.7-Sonnet | $7.50 \pm 0.50$ | $7.20 \pm 0.75$ | $2.80 \pm 1.99$ | $4.40 \pm 2.29$ | $3.50 \pm 2.29$ |

Table 16: End-to-end evaluation results judged by Claude-3.7-Sonnet-20250219.

| Model | Clarity | Novelty | Soundness | Significance | Overall |
|---|---|---|---|---|---|
| o4-mini-high | $8.00 \pm 0.00$ | $7.00 \pm 0.00$ | $5.60 \pm 0.66$ | $6.60 \pm 0.66$ | $5.70 \pm 0.78$ |
| Gemini-2.5-Pro-Preview | $7.80 \pm 0.40$ | $6.70 \pm 0.46$ | $4.70 \pm 0.78$ | $5.90 \pm 0.54$ | $5.20 \pm 0.87$ |
| Claude-3.7-Sonnet | $8.00 \pm 0.45$ | $7.00 \pm 0.45$ | $5.30 \pm 1.00$ | $6.60 \pm 0.80$ | $5.90 \pm 1.14$ |

Table 17: End-to-end performance comparison of AI Scientist V2 and MLR-Agent judged by Gemini-2.5-Pro-Preview-05-06. Each score is averaged across 10 tasks.

| Agent Scaffold | Review Dimensions | | | | |
|---|---|---|---|---|---|
| | Clarity | Novelty | Soundness | Significance | Overall |
| AI Scientist V2 | $6.50 \pm 1.69$ | $6.90 \pm 0.83$ | $5.00 \pm 2.37$ | $5.50 \pm 1.75$ | $5.00 \pm 2.19$ |
| MLR-Agent | $7.30 \pm 0.64$ | $6.80 \pm 0.60$ | $2.00 \pm 1.00$ | $3.50 \pm 0.92$ | $2.20 \pm 1.17$ |

Table 18: End-to-end performance comparison of AI Scientist V2 and MLR-Agent judged by Claude-3.7-Sonnet-20250219. Each score is averaged across 10 tasks.

| Agent Scaffold | Review Dimensions | | | | |
|---|---|---|---|---|---|
| | Clarity | Novelty | Soundness | Significance | Overall |
| AI Scientist V2 | $6.60 \pm 0.80$ | $6.50 \pm 0.50$ | $2.40 \pm 1.02$ | $4.20 \pm 1.25$ | $3.50 \pm 1.20$ |
| MLR-Agent | $8.00 \pm 0.00$ | $7.00 \pm 0.00$ | $5.60 \pm 0.66$ | $6.60 \pm 0.66$ | $5.70 \pm 0.78$ |

## D.1 Prompts for MLR-Agent

**MLR-Agent** is a simple and flexible scaffold designed to support open-ended research. It is implemented to favour simplicity over extensive prompt engineering, allowing us to assess each large language model's fundamental performance directly.

The automated research process includes four stages: (1) Idea Generation, (2) Proposal Generation, (3) Experimentation, and (4) Paper Writing. Since most models lack web access, we insert a literature review step using `GPT-4o-Search-Preview` [22] between stages (1) and (2), providing relevant reference material to the agent.

Below are the prompts used for each stage:

- Table 19 presents the prompt used for idea generation.
- Table 20 presents the prompt for instructing the model to perform a literature review.
- Table 21 presents the prompt used for generating the research proposal.
- Table 22 presents the prompt used for conducting experiments and obtaining results.
- Table 23 presents the prompt used for writing the research paper based on previously generated outputs, including the idea, proposal, and experimental results.

Table 19: **Prompt used for idea generation**. Users provide a task description (in this example, the introduction from the Building Trust workshop), which is combined with the prompt to instruct the backbone model (e.g., Claude) to generate a corresponding research idea.

| | |
|---|---|
| Task Input | # Workshop on Building Trust in Language Models and Applications
As Large Language Models (LLMs) are rapidly adopted across diverse industries, concerns around their trustworthiness, safety, and ethical implications increasingly motivate academic research, industrial development, and legal innovation...This workshop addresses the unique challenges posed by the deployment of LLMs, ranging from guardrails to explainability to regulation and beyond. The proposed workshop will bring together researchers and practitioners from academia and industry to explore cutting-edge solutions for improving the trustworthiness of LLMs and LLM-driven applications. The workshop will feature invited talks, a panel discussion, interactive breakout discussion sessions, and poster presentations, fostering rich dialogue and knowledge exchange. We aim to bridge the gap between foundational research and the practical challenges of deploying LLMs in trustworthy, use-centric systems.

## Workshop Scope

This workshop has a broad focus, including but not limited to:
1. Metrics, benchmarks, and evaluation of trustworthy LLMs
2. Improving reliability and truthfulness of LLMs
3. Explainability and interpretability of language model responses
...
7. Guardrails and regulations for LLMs
8. Error detection and correction |
| Prompt | You are an excellent machine learning researcher. Please generate innovative and practical ideas based on a given task description.
Note that there might be a couple of research topics in the task description, and you should focus on one of them.
The idea should be no more than 200 words and include the following three sections:
1. Title: A concise and descriptive title for the research idea.
2. Motivation: A brief explanation of why this research is important and what problems it aims to solve.
3. Main Idea: A clear and detailed description of the proposed research idea, including the methodology, expected outcomes, and potential impact.
Please directly respond to the idea.
Here is the task: {Task} |

| Output | # TrustPath: A Framework for Transparent Error Detection and Correction in LLMs |
|---|---|
| | ## Motivation |
| | As Large Language Models (LLMs) are increasingly integrated into high-stakes applications, their tendency to hallucinate, present false information, or make errors poses significant risks. Current error detection methods often lack transparency, making it difficult for users to understand when and why an LLM might be incorrect. This research addresses the critical need for systems that can not only detect errors in LLM outputs but also explain these errors to users and suggest corrections in a transparent, understandable manner. |
| | |
| | ## Main Idea |
| | TrustPath proposes a multi-layered approach to error detection and correction that prioritizes transparency. The framework employs three complementary mechanisms: 1) A self-verification module that prompts the LLM to evaluate its own uncertainty through generated alternative responses and explanations; 2) A factual consistency checker that retrieves information from trusted knowledge sources to verify claims; and 3) A human-in-the-loop feedback system that learns from user corrections. Uniquely, TrustPath presents detected errors alongside confidence scores, reasoning paths, and suggested alternatives through an intuitive visual interface that highlights potential problematic sections of text. This approach not only improves accuracy but builds user trust by making the error detection process visible and understandable, allowing users to make informed decisions about accepting or rejecting LLM outputs. |

Table 20: **Prompt used for literature review**. After generating the research idea, we use `GPT-4o-Search-Preview` [22] to conduct a literature review. The retrieved references are then provided to the model to guide the generation of the research proposal.

| Prompt | You are an excellent machine learning researcher! |
|---|---|
| | Please help me do a literature review for a given idea. |
| | The idea is based on a given task description. |
| | The papers in the literature review should be extracted from arxiv and published between 2023 to 2025. |
| | The literature review should include the following: |
| |   1. Related Papers: at least 10 academic papers most closely related to the current research idea, with a brief summary and the publication year of each one, organized logically. |
| |   2. Key Challenges: A discussion of the main challenges and limitations in the current research. List no more than five key challenges. |
| | The paper should be in the format of: |
| | 1. **Title**: <title> (<arxiv_id>) |
| |   - **Authors**: <author1>, <author2>, ... |
| |   - **Summary**: <summary> |
| |   - **Year**: <year> |
| | Please directly respond to the literature review and do not include any additional comments. |
| | Here is the idea: {Idea} |
| | Here is the task: {Task} |
| Output | 1. **Title**: Explainability for Large Language Models: A Survey (arXiv:2309.01029) |
| |   - **Authors**: Haiyan Zhao, Hanjie Chen, Fan Yang, Ninghao Liu, Huiqi Deng, Hengyi Cai, Shuaiqiang Wang, Dawei Yin, Mengnan Du |
| |   - **Summary**: This survey introduces a taxonomy of explainability techniques for Transformer-based language models, categorizing methods based on training paradigms and summarizing approaches for generating local and global explanations. It also discusses evaluation metrics and the use of explanations to debug models and improve performance. |
| |   - **Year**: 2023 |
| | |
| | ... |

10. **Title**: An Explainable AI Approach to Large Language Model Assisted Causal Model Auditing and Development (arXiv:2312.16211)
 - **Authors**: Yanming Zhang, Brette Fitzgibbon, Dino Garofolo, Akshith Kota, Eric Papenhausen, Klaus Mueller
 - **Summary**: This study proposes the use of LLMs as auditors for causal networks, presenting them with causal relationships to produce insights about edge directionality, confounders, and mediating variables. It envisions a system where LLMs, automated causal inference, and human analysts collaborate to derive comprehensive causal models.
 - **Year**: 2023

**Key Challenges:**

1. **Transparency and Interpretability**: LLMs often operate as black boxes, making it difficult to understand their internal decision-making processes. This lack of transparency hinders trust and accountability in high-stakes applications.

....

5. **User Feedback Mechanisms**: Incorporating human-in-the-loop feedback systems to learn from user corrections and improve model performance requires designing interfaces and processes that are both effective and user-friendly.

Table 21: **Prompt used for research proposal generation.** We provide the research task description, the previously generated research idea, and the literature review as inputs, which are combined with this prompt to guide the language model (e.g., Claude) in generating a complete research proposal.

| Prompt | You are an excellent machine learning researcher! Please generate a detailed research proposal based on a given task description, a research idea and their literature review. The proposal should be about 2000 words and include the following five sections: 1. Title: a concise and descriptive title for the research proposal. 2. Introduction: background, research objectives and significance. 3. Methodology: detailed and precise research design (including data collection, full algorithmic steps and/or mathematical formulas where appropriate, and full details about experimental design to validate the method, with evaluation metrics). 4. Expected Outcomes & Impact. The proposal should be well-structured and clearly articulate the research plan. When writing mathematical formulas, you should use LaTeX syntax. For inline formulas, use single dollar signs, for example: `$x^2$` to represent x squared. For block equations, use double dollar signs at the beginning and end, for example: `$$x^2$$`. Please directly respond to the proposal. Here is the task: {Task} Here is the idea: {Idea} Here is the literature review: {Related Work} |
| --- | --- |

Table 22: **Prompt used for experimentation**. We provide the research task description, the previously generated research idea, the literature review, and the research proposal as inputs, which are combined with this prompt to guide the coding agent (e.g., Claude) for conducting the experiments and obtaining the results.

| Prompt | Based on task.md, idea.md, related_work.md and proposal.md in this folder, please design an experimental plan to test the hypothesis that the proposed method is effective. Then write code to implement the experimental plan, and run the experiments in a fully automated manner. The experimental process should be fully automated and include the following steps: |
| --- | --- |

1. Create a folder named 'claude_code' inside this folder to save the code and results.
2. Write some Python scripts to run the experiment automatically. IMPORTANT: The script must be thoroughly debugged and tested until it can run successfully without any errors. The script should handle all necessary data processing, model training, and evaluation steps automatically. NOTE: If the environment has available GPUs, the script should utilize GPU acceleration where possible.
  - The scripts should be able to run all baseline methods automatically. - The scripts should be able to save the results in a structured format (e.g., CSV or JSON) for easy analysis.
  - The scripts should generate figures to visualize the results. Figures should be generated for the main results, including but not limited to:
  - Training and validation loss curves
  - Performance metrics (e.g., accuracy, F1 score) over time
  - Comparison of the proposed method with baseline methods
  - Any other relevant figures that help to understand the performance of the proposed method
  - Make sure that the figures are properly labeled and include legends, titles, and axis labels. There should be data points in the figures, and the figures should be easy to read and interpret.
3. Write a README.md file to explain how to run the experiment.
4. Run the experiment automatically, visualize the results, and save the results. Make sure the generated figure files are not empty.
5. Save the experiment execution process in log.txt.
6. Analyze and summarize the experiment results in results.md after all experiments (including all baselines) have been successfully completed.
  - Tables should be generated for the main results, including but not limited to:
  - Summary of the experimental setup (e.g., hyperparameters, dataset splits)
  - Comparison of the proposed method with baseline methods
  - Any other relevant tables that help to understand the performance of the proposed method
7. Finally, create a folder named 'results' under this folder and move results.md, log.txt, and all figures generated by the experiment into 'results'.

IMPORTANT:
  - Do not use synthetic results or generate any fake data. The results should be based on real experiments. If the experimental dataset is too large, please use a smaller subset of the dataset for testing purposes.
  - If the experiment is about large language models (LLMs), please confirm the models used are LLMs and not simple multi-layer neural networks such as LSTM.
  - You can download the datasets from Hugging Face datasets or other open-sourced datasets. Please do not use any closed-sourced datasets.
  - If the experiment requires using open-sourced large language models (LLMs) such as Meta Llama-3.1-8B, Llama-3.2-1B, Mistral Ministral-8B or Qwen Qwen3-0.6B and multimodal LLMs like Qwen2-VL-3B, please download the models from the Hugging Face model hub. Due to limited computing resources, please use the models no larger than 8B.
  -If the experiment requires using closed-sourced large language models (LLMs) such as OpenAI's GPT-4o-mini, o4-mini or Claude-3.7-sonnet, please directly use the API provided by the model provider. API keys have already been provided in the environment variables. Please use the API key to access the model and run the experiment.
  -Only save essential files that are necessary for the experiment. Do not create or save any unnecessary files, temporary files, or intermediate files. Keep the output minimal and focused on the core experiment requirements.
  -No need to generate any paper or academic document. Focus only on the experimental implementation and results.
  -Please remove checkpoints or datasets larger than 1MB after the experiment is completed.

Remember to analyze and summarize the experiment results, create a folder named 'results' under this folder and move results.md, log.txt, and all figures generated by the experiment into 'results'.
 - Figures and tables generated by the experiment should be included in the results.md with clear explanations of what they show. Please make sure the generated figures are not repeated. Please do not use the same figures multiple times in results.md.
 - Make sure paths to the figures in results.md are correct and point to the right locations.
 - The results.md should also include a discussion of the results, including any insights gained from the experiment and how they relate to the hypothesis.
 - The results.md should include a summary of the main findings and conclusions drawn from the experiment.
 - The results.md should also include any limitations of the experiment and suggestions for future work.

Table 23: **Prompt used for paper writing.** We compile all outputs from the previous stages and combine them with the following prompt to instruct the LLM to generate a complete research paper.

| Prompt | You are an excellent machine learning researcher! |
|---|---|
| | Given the task, research idea, literature review, proposal and experiment results, please write a paper for the machine learning project. |
| | It should include the following sections: |
| | 1. Title and Abstract: A concise title and a brief abstract summarizing the research. |
| | 2. Introduction: An introduction to the problem, its significance, and the proposed solution. |
| | 3. Related Work: A review of existing literature and how your work fits into the current landscape. |
| | 4. Methodology: A detailed description of the proposed method, including any algorithms or models used. |
| | 5. Experiment Setup: A description of the experimental setup, including datasets, metrics, and evaluation methods. |
| | 6. Experiment Results: A presentation of the results obtained from the experiments, including tables and figures. |
| | 7. Analysis: An analysis of the results, discussing their implications and any limitations. |
| | 8. Conclusion: A summary of the findings and suggestions for future work. |
| | 9. References: A list of references cited in the paper. |
| | |
| | Figures and tables should be included in the paper. You may refer to the figures and tables in the experiment results. If there is no image in the experiment results, please do not create or cite any fake figures. Please directly use the paths of the figures in the markdown file and do not use any placeholders. |
| | When writing mathematical formulas, you should use LaTeX syntax. For inline formulas, use single dollar signs, for example: $x^2$ to represent x squared. For block equations, use double dollar signs at the beginning and end, for example: $x^2$. |
| | If you need to write text in mathematical formulas, please avoid invalid escape characters. |
| | The paper should be well-structured and clearly present the research findings. |
| | Here is the task: {Task} |
| | Here is the idea: {Idea} |
| | Here is the literature review: {Related Work} |
| | Here is a summary of the experiment results: {Experiment Results} |

## D.2 Prompts and Rubrics for MLR-Judge

**MLR-Judge** combines a set of carefully designed review rubrics with an LLM-based evaluator. It supports both an end-to-end evaluation pipeline—assessing an AI agent's ability to complete a full research project—and a stepwise evaluation pipeline that independently evaluates performance on (1) Idea Generation, (2) Proposal Generation, (3) Experimentation, and (4) Paper Writing. In the

following sections, we describe the specific rubrics used for each stage as well as for end-to-end evaluation. We then present the prompt used to operate the MLR-Judge.

### D.2.1 Review Rubrics

We provide the rubrics used to guide evaluation for each stage:

- The rubrics used to evaluate the idea generation quality are shown in Table 26.
- The rubrics used to evaluate the research proposal quality are shown in Table 25.
- The rubrics used to evaluate the experimentation quality are shown in Table 26.
- The rubrics used to evaluate the paper writing quality are shown in Table 27.
- The rubric used for holistic paper evaluation is shown in Table 28.

Table 24: Rubrics used for idea generation.

1. **CONSISTENCY** (1-10)
 How well does the idea align with the requirements of the task description?

 9-10 - Excellent: The idea is perfectly aligned with the task description. It addresses all aspects of the task and is highly relevant.
 7-8 - Good: The idea is mostly aligned with the task description. It addresses most aspects but may miss some minor details.
 5-6 - Satisfactory: The idea is somewhat aligned with the task description. It addresses some aspects but misses several key points.
 3-4 - Needs Improvement: The idea is poorly aligned with the task description. It addresses only a few aspects and misses many key points.
 1-2 - Poor: The idea is not aligned with the task description. It does not address the task or is completely irrelevant.

2. **CLARITY** (1-10)
 How clear and well-defined is the research idea?

 9-10 - Excellent: The idea is crystal clear, perfectly defined, and immediately understandable. It is articulated concisely with no ambiguity.
 7-8 - Good: The idea is mostly clear and well-articulated with only minor ambiguities. Minor refinements would make it even more precise.
 5-6 - Satisfactory: The idea is partially clear but has some ambiguities. Some aspects need further elaboration for a complete understanding.
 3-4 - Needs Improvement: The idea is unclear with significant ambiguities. Major clarification is needed to make it comprehensible.
 1-2 - Poor: The idea is extremely unclear, vague, or ambiguous with no proper explanation. It is difficult to understand or interpret.

3. **NOVELTY** (1-10)
 How original and innovative is the research idea?

 9-10 - Excellent: The idea is highly original and innovative. It introduces a groundbreaking concept or approach that is significantly different from existing work.
 7-8 - Good: The idea has notable originality and innovation. It offers fresh perspectives or new combinations of existing concepts.
 5-6 - Satisfactory: The idea has some originality and innovation. It includes some novel aspects but also shares similarities with existing approaches.
 3-4 - Needs Improvement: The idea has minimal originality. It largely resembles existing approaches with only slight variations.
 1-2 - Poor: The idea lacks originality and innovation. It closely resembles common or existing concepts without any new insights.

4. **FEASIBILITY** (1-10)

How practical and implementable is the research idea?

9-10 - Excellent: The idea is highly practical and implementable with current resources, technology, and knowledge. Execution is straightforward.
7-8 - Good: The idea is largely feasible with existing technology and methods, though it may require moderate refinement or optimization.
5-6 - Satisfactory: The idea is somewhat feasible but has some implementation challenges. It may require considerable effort or resources to implement.
3-4 - Needs Improvement: The idea has significant implementation challenges. Major revisions would be needed to make it feasible.
1-2 - Poor: The idea is impractical or impossible to implement with current technology, knowledge, or constraints.

5. **SIGNIFICANCE** (1-10)
How important and impactful is the research idea?

9-10 - Excellent: The idea is highly significant and impactful. It addresses a critical problem and could lead to major advancements in the field.
7-8 - Good: The idea is significant and has clear impact potential. It addresses an important issue and could lead to meaningful contributions.
5-6 - Satisfactory: The idea is somewhat significant. It addresses a relevant problem but its impact may be moderate or limited to a specific area.
3-4 - Needs Improvement: The idea has limited significance. It addresses a minor issue or has a narrow scope with minimal impact.
1-2 - Poor: The idea has little to no significance. It does not address a meaningful problem or offer any clear benefits.

6. **Overall Assessment** (1-10)
How would you rate the research idea overall, considering all five dimensions above?

10 - Outstanding: The idea is exceptional in every respect, with no significant weaknesses.
8-9 - Excellent: The idea is very strong overall, with only minor weaknesses.
6-7 - Good: The idea is solid, with a good balance of strengths and weaknesses.
4-5 - Satisfactory: The idea is adequate but has notable weaknesses.
2-3 - Needs Improvement: The idea has significant weaknesses that limit its potential.
1 - Poor: The idea is fundamentally flawed across most or all dimensions.

When assigning the Overall Assessment score, consider not just the average of the five dimensions, but also:
- Whether any single weakness is critical enough to lower the overall potential.
- The overall coherence and integration of the idea.
- The likelihood of real-world impact if the idea were pursued.
- The degree to which the idea fulfills the task description as a whole.
- Any unique strengths or fatal flaws that are not fully captured by the individual dimensions.

Table 25: Rubrics used for proposal generation.

1. **CONSISTENCY** (1-10)
How well does the proposal align with the requirements of the task description, the research idea, and the literature review?

9-10 - Excellent: The proposal is perfectly aligned with the task description, research idea, and literature review. It addresses all aspects comprehensively and demonstrates a deep understanding of the context and prior work. There are no inconsistencies or gaps.
7-8 - Good: The proposal is mostly aligned with the task description, research idea, and literature review. It addresses most aspects, with only minor omissions or inconsistencies that do not significantly affect the overall coherence.

5-6 - Satisfactory: The proposal is somewhat aligned. It addresses some key aspects of the task description, research idea, and literature review, but misses several important points or contains moderate inconsistencies.

3-4 - Needs Improvement: The proposal is poorly aligned. It addresses only a few aspects of the task description, research idea, or literature review, and contains significant inconsistencies or gaps that undermine its coherence.

1-2 - Poor: The proposal is not aligned with the task description, research idea, or literature review. It fails to address the requirements or is completely irrelevant, with major inconsistencies or contradictions.

## 2. **CLARITY** (1-10)
How clear and well-defined is the research proposal?

9-10 - Excellent: The proposal is crystal clear, perfectly defined, and immediately understandable. All objectives, methods, and rationales are articulated concisely with no ambiguity. The structure is logical and easy to follow.

7-8 - Good: The proposal is mostly clear and well-articulated, with only minor ambiguities or areas that could benefit from slight refinement. The main points are understandable and the structure is generally logical.

5-6 - Satisfactory: The proposal is partially clear but has some ambiguities or unclear sections. Some aspects require further elaboration or clarification for a complete understanding. The structure may be somewhat disjointed.

3-4 - Needs Improvement: The proposal is unclear with significant ambiguities or confusing sections. Major clarification is needed to make the objectives, methods, or rationale comprehensible. The structure is difficult to follow.

1-2 - Poor: The proposal is extremely unclear, vague, or ambiguous with no proper explanation. It is difficult to understand or interpret, and the structure is disorganized or incoherent.

## 3. **NOVELTY** (1-10)
How original and innovative is the research proposal?

9-10 - Excellent: The proposal is highly original and innovative. It introduces a groundbreaking concept, method, or perspective that is significantly different from existing work in the literature. The novelty is clearly articulated and well-justified.

7-8 - Good: The proposal demonstrates notable originality and innovation. It offers fresh perspectives or new combinations of existing concepts, with clear distinctions from prior work, though it may not be entirely groundbreaking.

5-6 - Satisfactory: The proposal has some originality and innovation. It includes novel aspects but also shares similarities with existing approaches. The differences from prior work are present but not strongly emphasized.

3-4 - Needs Improvement: The proposal has minimal originality. It largely resembles existing approaches in the literature, with only slight variations or incremental improvements.

1-2 - Poor: The proposal lacks originality and innovation. It closely follows common or existing concepts without any new insights, and does not distinguish itself from prior work.

## 4. **SOUNDNESS** (1-10)
How well-founded and rigorous is the research proposal?

9-10 - Excellent: The proposal is highly sound and rigorous. It is based on solid theoretical foundations, well-established methods, and comprehensive literature review. The proposed methodology is robust and well-justified. Technical formulations are fully correct and clearly presented.

7-8 - Good: The proposal is sound and mostly rigorous. It is based on solid foundations and established methods, though it may have minor gaps or areas that require further justification. The methodology is generally well-defined. Technical formulations are mostly correct, with minor errors or omissions.

5-6 - Satisfactory: The proposal is somewhat sound but has some gaps or weaknesses in its theoretical foundations or methodology. It may rely on assumptions that are not fully justified. The proposed methods are acceptable but may lack rigor. Technical formulations have some errors or unclear aspects.

3-4 - Needs Improvement: The proposal has significant weaknesses in its soundness or rigor. It may rely on questionable assumptions, poorly defined methods, or lack sufficient justification for its approach. Technical formulations are often incorrect or poorly presented.

1-2 - Poor: The proposal is fundamentally unsound or lacks rigor. It is based on flawed assumptions, poorly defined methods, or lacks sufficient justification for its approach. Technical formulations are incorrect or absent.

## 5. FEASIBILITY (1-10)
How practical and implementable is the research proposal?

9-10 - Excellent: The proposal is highly practical and implementable with current resources, technology, and knowledge. The plan is realistic, and execution is straightforward with clearly defined steps and minimal risk.

7-8 - Good: The proposal is largely feasible with existing technology and methods, though it may require moderate refinement, optimization, or additional resources. The plan is generally realistic, with manageable risks.

5-6 - Satisfactory: The proposal is somewhat feasible but presents some implementation challenges. It may require considerable effort, resources, or further development to implement successfully. Some risks or uncertainties are present.

3-4 - Needs Improvement: The proposal has significant implementation challenges. Major revisions, additional resources, or new methods would be needed to make it feasible. There are substantial risks or uncertainties that threaten successful execution.

1-2 - Poor: The proposal is impractical or impossible to implement with current technology, knowledge, or constraints. The plan is unrealistic, and the likelihood of successful execution is extremely low.

## 6. SIGNIFICANCE (1-10)
How important and impactful is the research proposal?

9-10 - Excellent: The proposal is highly significant and impactful. It addresses a critical problem or gap in the field and has the potential to lead to major advancements or transformative change. The expected contributions are substantial and clearly articulated.

7-8 - Good: The proposal is significant and has clear impact potential. It addresses an important issue and could lead to meaningful contributions or improvements in the field, though the impact may not be transformative.

5-6 - Satisfactory: The proposal is somewhat significant. It addresses a relevant problem, but its impact may be moderate or limited to a specific area or community. The expected contributions are present but not far-reaching.

3-4 - Needs Improvement: The proposal has limited significance. It addresses a minor issue or has a narrow scope, with minimal potential for impact or advancement in the field.

1-2 - Poor: The proposal has little to no significance. It does not address a meaningful problem or offer any clear benefits, and its potential impact is negligible or absent.

## 7. Overall Assessment (1-10)
How would you rate the research proposal overall, considering all five dimensions above?

10 - Outstanding: The proposal is exceptional in every respect, with no significant weaknesses. It demonstrates excellence across all dimensions and has the potential for major impact.

8-9 - Excellent: The proposal is very strong overall, with only minor weaknesses. It is well-balanced, highly promising, and likely to make a significant contribution.

6-7 - Good: The proposal is solid, with a good balance of strengths and weaknesses. It is generally well-conceived and feasible, though some areas could be improved.

4-5 - Satisfactory: The proposal is adequate but has notable weaknesses that limit its potential. It addresses the main requirements but lacks strength in one or more key areas.

2-3 - Needs Improvement: The proposal has significant weaknesses that limit its potential for success or impact. Major revisions are needed to address critical issues.

1 - Poor: The proposal is fundamentally flawed across most or all dimensions. It is unlikely to succeed or make a meaningful contribution without substantial reworking.

When assigning the Overall Assessment score, consider not just the average of the six dimensions, but also:
- Whether any single weakness is critical enough to lower the overall potential.
- The overall coherence and integration of the proposal.
- The likelihood of real-world impact if the proposal were pursued.
- The degree to which the proposal fulfills the task description, research idea, and literature review as a whole.
- Any unique strengths or fatal flaws that are not fully captured by the individual dimensions.

Table 26: Rubrics used for experimentation.

1. **Hallucination** (True/False)
Does the experimental document contain any hallucinated content? Hallucinated content refers to information that is fabricated or incorrect, and does not align with the provided task description, research idea, literature review, or research proposal. Fake data, results, or methods should be considered as hallucinated content.

True - The experimental document contains hallucinated content.
False - The experimental document does not contain any hallucinated content.

2. **Consistency** (1-10)
How well do the experimental document align with the task description, research idea, literature review and research proposal? Does the implementation match the proposed methods and ideas?

9-10 - Excellent: The experimental document are fully consistent with the task description, research idea, literature review and research proposal. There are no discrepancies or contradictions. The implementation is a perfect match to the proposed methods and ideas.
7-8 - Good: The experimental document are mostly consistent, with only minor discrepancies or contradictions. The main points are well-aligned with the task, idea, literature review and proposal. The implementation is largely aligned with the proposed methods and ideas.
5-6 - Moderate: Some inconsistencies or unclear alignments exist, but overall the experimental document are still relevant to the task, idea, literature review and proposal. The implementation is somewhat aligned with the proposed methods and ideas.
3-4 - Weak: Significant inconsistencies or contradictions are present, leading to confusion or misalignment with the task, idea, literature review and proposal. The implementation is poorly aligned with the proposed methods and ideas.
1-2 - Poor: The experimental document are largely inconsistent with the task, idea, literature review and proposal, or there are major contradictions. The implementation is not aligned with the proposed methods and ideas at all.

3. **Completeness** (1-10)
Are all necessary experiments, baselines, and ablation studies included in this experimental document? Are all relevant results reported, and is the experimental setup fully described?

9-10 - Excellent: All necessary experiments, baselines, and ablations are included. The results are comprehensive and the setup is fully described.
7-8 - Good: Most necessary experiments, baselines, and ablations are included, with only minor omissions. The setup is mostly described.
5-6 - Moderate: Some important experiments, baselines, and ablations are missing, but the main points are covered. The setup is partially described.
3-4 - Weak: Many key experiments are missing, making the evaluation incomplete. The setup is poorly described and lacks clarity.
1-2 - Poor: The results are highly incomplete, with major omissions. The setup is missing.

4. **Novelty** (1-10)
Does the experiment document demonstrate new findings, methods, or insights compared to existing work? Is the experimental design innovative or derivative?

9-10 - Excellent: The experimental document presents highly novel findings, methods, or insights that significantly advance the field. The design is innovative and original.

7-8 - Good: The experimental document shows some novel aspects, with a good level of innovation. The design is mostly original.

5-6 - Moderate: The experimental document has some novel elements, but they are not particularly groundbreaking. The design is somewhat derivative.

3-4 - Weak: The experimental document lacks novelty, with little to no new findings or insights. The design is largely derivative.

1-2 - Poor: The experimental document is entirely derivative, with no new findings or insights. The design is unoriginal and lacks creativity.

5. **Soundness** (1-10)
Are the experimental methods, analysis, and conclusions logically sound and scientifically rigorous? Are the results reproducible and well-supported?

9-10 - Excellent: Methods and analysis are highly rigorous, logically sound, and statistically valid. Results are fully reproducible and well-supported.

7-8 - Good: Methods and analysis are generally sound, with only minor issues. Results are mostly reproducible and well-supported.

5-6 - Moderate: Some weaknesses in rigor or logic, but the main conclusions are still supported. Results are partially reproducible.

3-4 - Weak: Significant flaws in methods or analysis, casting doubt on the conclusions. Results are not well-supported or reproducible.

1-2 - Poor: Methods or analysis are fundamentally unsound or unscientific. Results are not reproducible and conclusions are unsupported.

6. **Insightfulness** (1-10)
Do the results provide deep insights, meaningful interpretations, or valuable implications for the field? Are trends, patterns, and implications discussed thoughtfully?

9-10 - Excellent: Results are analyzed in depth, with highly insightful interpretations and valuable implications for the field.

7-8 - Good: Results are thoughtfully analyzed, with some meaningful insights.

5-6 - Moderate: Some insights are provided, but analysis is relatively superficial.

3-4 - Weak: Little meaningful analysis or interpretation is provided.

1-2 - Poor: No insight or thoughtful analysis is present.

7. **Significance** (1-10)
How important or impactful are the experiment results for the field? Do they address a critical problem or open new research directions?

9-10 - Excellent: Results are highly significant, addressing critical problems or opening important new directions.

7-8 - Good: Results are significant and make a clear contribution.

5-6 - Moderate: Results are somewhat significant, but impact is limited.

3-4 - Weak: Results have little significance or impact.

1-2 - Poor: Results are insignificant or irrelevant.

8. **Overall Assessment** (1-10)
Provide an overall assessment of the experimental work, considering all the above dimensions.

10 - Outstanding: The experimental work is exemplary in every respect, demonstrating exceptional quality, rigor, and impact. All aspects are handled with great care and expertise, with no significant weaknesses. The work sets a high standard and is likely to have a major influence in its field.

8-9 - Excellent: The work is very strong overall, with clear strengths across most or all dimensions. Any weaknesses are minor and do not detract from the overall quality or credibility. The experimental work is well-executed, insightful, and makes a significant contribution.

6-7 - Good: The experimental work is solid and generally well-conceived, with a good balance of strengths and weaknesses. While there may be some areas for improvement, the work is credible, meaningful, and meets the main requirements for quality research.

4-5 - Satisfactory: The work is adequate but has several notable weaknesses that limit its overall quality or impact. While it addresses the main objectives, shortcomings in design, execution, or analysis reduce its effectiveness and significance.

2-3 - Needs Improvement: The experimental work has substantial weaknesses in multiple areas, which undermine its credibility or value. Major revisions and improvements are needed for the work to reach an acceptable standard.

1 - Poor: The work is fundamentally flawed across most or all dimensions. It fails to meet essential standards for research quality and is unlikely to provide meaningful insights or contributions.

When assigning the Overall Assessment score, consider not just the average of these dimensions, but also:
- The overall coherence and integration of the experimental work.
- The presence of any particularly outstanding strengths or critical weaknesses that may not be fully reflected in the individual scores.
- The potential impact or importance of the work as a whole.
- The degree to which the experimental work advances the field or opens new research directions.
- Any unique contributions, innovative aspects, or serious flaws that significantly affect the overall quality.
Your overall assessment should reflect a holistic judgment, taking into account both the quantitative scores and your qualitative evaluation of the experimental work.

Table 27: Rubrics used for paper writing.

1. **Consistency** (1-10)
 - How well does the paper align with the task description, research idea, research proposal and experimental results?
 - Are there any contradictions or inconsistencies in the paper's arguments or findings?

 9-10 - Excellent: The paper is highly consistent, with no contradictions or inconsistencies.
 7-8 - Good: The paper is mostly consistent, with minor contradictions or inconsistencies.
 5-6 - Fair: The paper has some inconsistencies, but they do not significantly detract from the overall quality.
 3-4 - Poor: The paper has several inconsistencies that detract from the overall quality.
 1-2 - Very Poor: The paper is highly inconsistent, with major contradictions or inconsistencies that significantly detract from the overall quality.

2. **Clarity** (1-10)
 - How clear and understandable is the paper's writing?
 - Are the arguments and findings presented in a logical and coherent manner?
 - Is the paper well-structured and easy to follow?

 9-10 - Excellent: The paper is very clear and easy to understand, with a logical structure and coherent arguments.
 7-8 - Good: The paper is mostly clear, with minor issues in structure or coherence.
 5-6 - Fair: The paper has some clarity issues, but they do not significantly detract from the overall quality.
 3-4 - Poor: The paper has several clarity issues that detract from the overall quality.
 1-2 - Very Poor: The paper is very unclear and difficult to understand, with major issues in structure or coherence.

3. **Completeness** (1-10)
 - How complete is the paper in terms of addressing the task description, research idea, research proposal and experimental results?
 - Are there any missing components or sections that should be included?

9-10 - Excellent: The paper is very complete, addressing all components of the task description, research idea, research proposal and experimental results.
7-8 - Good: The paper is mostly complete, with minor missing components or sections.
5-6 - Fair: The paper has some missing components or sections, but they do not significantly detract from the overall quality.
3-4 - Poor: The paper has several missing components or sections that detract from the overall quality.
1-2 - Very Poor: The paper is very incomplete, with major missing components or sections that significantly detract from the overall quality.

4. **Soundness** (1-10)
 - Are the arguments and findings supported by evidence and reasoning?
 - Are there any flaws or weaknesses in the paper's methodology or approach?
 - Are the experimental results and analyses valid and reliable?

9-10 - Excellent: The paper is very sound, with strong evidence and reasoning supporting the arguments and findings. The experimental results and analyses are fully valid and reliable.
7-8 - Good: The paper is mostly sound, with minor flaws or weaknesses in methodology or approach. The experimental results and analyses are mostly valid and reliable.
5-6 - Fair: The paper has some flaws or weaknesses in methodology or approach, but they do not significantly detract from the overall quality. The experimental results and analyses are somewhat valid and reliable.
3-4 - Poor: The paper has several flaws or weaknesses in methodology or approach that detract from the overall quality. Many of the experimental results and analyses are not valid or reliable.
1-2 - Very Poor: The paper is very unsound, with major flaws or weaknesses in methodology or approach that significantly detract from the overall quality. The experimental results and analyses are not valid or reliable.

5. **Overall Assessment** (1-10)
 - Based on your evaluation of the paper's consistency, clarity, completeness and soundness, what is your overall assessment of the paper?

10 - Outstanding: The paper is of outstanding quality, with no issues in any of the key dimensions.
8-9 - Excellent: The paper is of excellent quality, with minor issues in one or more of the key dimensions.
6-7 - Good: The paper is of good quality, with some issues in one or more of the key dimensions.
4-5 - Fair: The paper is of fair quality, with several issues in one or more of the key dimensions.
2-3 - Poor: The paper is of poor quality, with major issues in one or more of the key dimensions.
1 - Very Poor: The paper is of very poor quality, with major issues in all of the key dimensions.

When assigning the Overall Assessment score, consider not just the average of these dimensions, but also:
- The overall coherence and integration of the paper.
- The presence of any particularly outstanding strengths or critical weaknesses that may not be fully reflected in the individual scores.
- The potential impact or importance of the work as a whole.
- The degree to which the paper advances the field or opens new research directions.
- Any unique contributions, innovative aspects, or serious flaws that significantly affect the overall quality.
Your overall assessment should reflect a holistic judgment, taking into account both the quantitative scores and your qualitative evaluation of the paper.

Table 28: Prompt used for end-to-end evaluation.

1. **Clarity** (1-10)
 - Is the paper well-written and easy to understand?
 - Are the ideas and contributions clearly articulated?
 - Is the structure of the paper logical and coherent?

9-10 - The paper is exceptionally well-written, with clear and concise language. The ideas are presented in a logical and coherent manner, making it easy to follow the author's arguments.

7-8 - The paper is well-written, but there are some areas that could be improved for clarity. The ideas are mostly clear, but there may be some minor issues with the structure or language.

5-6 - The paper is somewhat difficult to read, with several areas that are unclear or poorly articulated. The structure may be confusing, making it hard to follow the author's arguments.

3-4 - The paper is poorly written, with many unclear or confusing sections. The ideas are not well-articulated, and the structure is disorganized.

1-2 - The paper is extremely difficult to read, with numerous unclear or confusing sections. The ideas are poorly articulated, and the structure is completely disorganized.

2. **Novelty** (1-10)
 - Does the paper present new and original ideas and findings?
 - Are the experimental results and contributions original and novel?
 - Is the work a significant advance over existing research?

9-10 - The paper presents groundbreaking ideas and findings that are highly original and significant. The contributions are a major advance over existing research and are likely to have a lasting impact on the field.

7-8 - The paper presents some new and original ideas, and the contributions are significant. The work is a notable advance over existing research, but it may not be as groundbreaking as top-tier papers.

5-6 - The paper presents some new ideas and findings, but they are not particularly original or significant. The contributions are somewhat incremental and do not represent a major advance over existing research.

3-4 - The paper presents few new ideas or findings, and those that are presented are not original or significant. The contributions are minimal and do not advance the field.

1-2 - The paper presents no new ideas, and the contributions are completely unoriginal. The work does not advance the field in any meaningful way.

3. **Soundness** (1-10)
 - Are the methods and techniques used in the paper sound and appropriate?
 - Are the results and conclusions supported by the data?
 - Are there any major flaws or weaknesses in the experimental design, results or analysis?
 - Are the experimental results reliable and consistent to the code of the paper? Are the experimental results real or fake?
 - Are the visualization and analysis figures based on real experimental results or based on fake data?

9-10 - The methods and techniques used in the paper are sound and appropriate. The results are well-supported by the data, and there are no major flaws or weaknesses in the experimental design, results or analysis. The experimental results are fully reliable and consistent with the code of the paper.

7-8 - The methods and techniques used in the paper are mostly sound, but there may be some minor issues. The results are generally well-supported by the data, but there may be some areas that could be improved. The experimental design, results or analysis may have some minor flaws. The experimental results are mostly reliable.

5-6 - The methods and techniques used in the paper are somewhat questionable, with several areas that could be improved. The results are not well-supported by the data, and there may be some significant flaws in the experimental design, results or analysis. Some experimental results are not reliable.

3-4 - The methods and techniques used in the paper are flawed or inappropriate. The results are not well-supported by the data, and there are major flaws in the experimental design, results or analysis. Most of experimental results are not reliable.

1-2 - The methods and techniques used in the paper are completely unsound. The results are not supported by the data, and there are numerous major flaws in the experimental design, results or analysis. The conclusions drawn from the paper are completely invalid. All experimental results are not reliable.

4. **Significance** (1-10)
 - Does the paper address an important problem or question?
 - Are the contributions significant to the field?
 - Are the experimental results reproducible and reliable? Do they have a significant impact?
 - Will the work have a lasting impact on the field?

 9-10 - The paper addresses a highly important problem or question, and the results and contributions are significant to the field. The work is likely to have a lasting impact on the field.
 7-8 - The paper addresses an important problem or question, and the results and contributions are significant. The work may have a lasting impact on the field, but it may not be as groundbreaking as top-tier papers.
 5-6 - The paper addresses a somewhat important problem or question, but the results and contributions are not particularly significant. The work may have some impact on the field, but it is unlikely to be lasting.
 3-4 - The paper addresses a minor problem or question, and the results and contributions are minimal. The work is unlikely to have any significant impact on the field.
 1-2 - The paper addresses an unimportant problem or question, and the results and contributions are completely insignificant. The work will have no impact on the field.

5. **Overall Assessment** (1-10)
 - Based on the above criteria, how would you rate the overall quality of the paper? Note that any single weakness can be critical to lower the overall assessment.
 - Is the paper suitable for publication in a top-tier conference or journal?
 - Would you recommend this paper to your colleagues?

 10 - The paper is of exceptional quality and is highly suitable for publication in a top-tier conference or journal. I would strongly recommend this paper.
 8-9 - The paper is of high quality and is suitable for publication in a top-tier conference or journal. I would recommend this paper.
 6-7 - The paper is of good quality and is suitable for publication in a reputable conference or journal. I would recommend this paper with some reservations.
 4-5 - The paper is of acceptable quality but may not be suitable for publication in a top-tier conference or journal. I would recommend this paper with significant reservations.
 2-3 - The paper is of poor quality and is not suitable for publication in a top-tier conference or journal. I would not recommend this paper.
 1 - The paper is of extremely poor quality and is not suitable for publication in any conference or journal. I would strongly advise against recommending this paper.

6. Confidence Score (1-5)
 - How confident are you in your overall assessment of the paper?

 5 - Extremely confident in the overall assessment.
 4 - Very confident in the overall assessment.
 3 - Moderately confident in the overall assessment.
 2 - Slightly confident in the overall assessment.
 1 - Not confident in the overall assessment.

Please provide a detailed review of the paper, including your scores for each aspect and an overall assessment. Be sure to justify your scores with specific examples from the paper.
Please do not include any personal opinions or biases in your review. Your review should be objective and based solely on the content of the paper. Please provide a confidence score from 1 to 5 for the overall assessment.
Do not hesitate to assign lower scores if the paper does not fully meet the criteria. Avoid giving high scores by default.

### D.2.2 Evaluation Prompts for MLR-Judge

These prompts instruct the model to apply the above rubrics and format its review output accordingly.

- Table 30 shows the prompt used for evaluating the overall quality of a research paper in an end-to-end manner.
- Table 29 shows the prompt used for evaluating the quality of the generated research idea.
- Table 33 shows the prompt used for evaluating the research proposal.
- Table 32 shows the prompt used for evaluating the experimental implementation and results.
- Table 31 shows the prompt used for evaluating the paper write-up.

Table 29: Instruction prompt used for evaluating the quality of generated idea. To ensure the model adheres to the desired output format, we specify the format twice in the prompt—once in the middle and again at the end. For brevity, this repeated instruction (i.e., at the end) is omitted here. Please refer to our code repository for full details.

---

You are an expert machine learning researcher!
You will be given a research idea and a task description.
Your task is to evaluate a research idea on a scale of 1 to 10 across five key dimensions and finally give an overall assessment on a scale of 1 to 10.
Please be objective in your evaluation, and provide detailed justifications for each score you assign. Do not hesitate to assign lower scores if the idea does not fully meet the criteria. Avoid giving high scores by default.

## Evaluation Rubric
{Idea Evaluation Rubrics as shown in Table 24}

## Output Format
Please output a complete JSON object strictly following the format below, including all evaluation items (Consistency, Clarity, Novelty, Feasibility, Significance, OverallAssessment). Do not output only a single item or partial content; you must output the entire JSON object.

```json
{
    "Consistency": {
        "score": <1-10>,
        "justification": "<detailed explanation of why the score
            was given, referencing the alignment with the task
            description>"
    },
    "Clarity": {
        "score": <1-10>,
        "justification": "<detailed explanation of why the score
            was given, referencing the clarity of the idea>"
    },
    "Novelty": {
        "score": <1-10>,
        "justification": "<detailed explanation of why the score
            was given, referencing the originality and innovation of
             the idea>"
    },
    "Feasibility": {
        "score": <1-10>,
        "justification": "<detailed explanation of why the score
            was given, referencing the practicality and
            implementability of the idea>"
    },
    "Significance": {
        "score": <1-10>,
        "justification": "<detailed explanation of why the score
            was given, referencing the importance and impact of the
            idea>"
    },
    "OverallAssessment": {
        "score": <1-10>,
        "strengths": ["<strength 1>", "<strength 2>"],
        "weaknesses": ["<weakness 1>", "<weakness 2>"]
    }
}
```

Please make sure your output is a complete JSON object and includes all the fields above.
Please make sure the answer is strictly in valid JSON format.
- Do not include any text, comments, or explanations outside the JSON code block.
- Do not use trailing commas.
- Do not use single quotes; use double quotes for all keys and string values.
- Do not include comments inside the JSON.
- Do not use unescaped control characters (e.g., newlines inside strings).
- Do not use unquoted keys; all keys must be in double quotes.
- Do not use invalid values like NaN or Infinity.
- Ensure all brackets and braces are properly closed.

Table 30: Instruction prompt used for evaluating the overall quality of the paper. To ensure the model adheres to the desired output format, we specify the format twice in the prompt—once in the middle and again at the end. For brevity, this repeated instruction (i.e., at the end) is omitted here. Please refer to our code repository for full details.

You are an expert machine learning researcher! You will be given a research paper which is based on a task description. You might also be given the code of the paper to check the reproducibility of the paper.
You task is to review the paper in terms of 4 key aspects - Clarity, Novelty, Soundness and Significance. Please provide a score from 1 to 10 for each aspect and an overall assessment, where 1 is the lowest and 10 is the highest. Lastly, provide a confidence score from 1 to 5 for the overall assessment, where 1 is the lowest and 10 is the highest.

## Evaluation Rubric
{Overall Evaluation Rubrics as shown in Table 28}

Please provide a detailed review of the paper, including your scores for each aspect and an overall assessment. Be sure to justify your scores with specific examples from the paper.
Please do not include any personal opinions or biases in your review. Your review should be objective and based solely on the content of the paper. Please provide a confidence score from 1 to 5 for the overall assessment. Do not hesitate to assign lower scores if the paper does not fully meet the criteria. Avoid giving high scores by default.

## Output Format

Please provide your review in the following format:

```
{
    "Clarity": {
        "score": <1-10>,
        "justification": "<Your justification here>"
    },
    "Novelty": {
        "score": <1-10>,
        "justification": "<Your justification here>"
    },
    "Soundness": {
        "score": <1-10>,
        "justification": "<Your justification here>"
    },
    "Significance": {
        "score": <1-10>,
        "justification": "<Your justification here>"
    },
    "Overall": {
        "score": <1-10>,
        "strengths": ["<strength 1>", "<strength 2>"],
        "weaknesses": ["<weakness 1>", "<weakness 2>"]
    },
    "Confidence": <1-5>
}
```

Note that any single weakness can be critical to lower the overall assessment.
Please provide detailed justifications for each score, including specific examples from the paper.
IMPORTANT: Please ensure that your output is a complete and valid JSON object and includes all the fields above. Do not output only a single item or partial content; you must output the entire JSON object.

## Task Description
{task}

## Paper to Be Reviewed
Note: The paper is generated by AI and may contain some errors. Please check the paper carefully and provide your review.
{paper}

## Code of the Paper
{code content}
Please provide a detailed review of the paper, including your scores for each aspect and an overall assessment. Be sure to justify your scores with specific examples from the paper.
Please do not include any personal opinions or biases in your review. Your review should be objective and based solely on the content of the paper. Please provide a confidence score from 1 to 5 for the overall assessment.

Do not hesitate to assign lower scores if the paper does not fully meet the criteria. Avoid giving high scores by default.

Table 31: Instruction prompt used for paper writing. To ensure the model adheres to the desired output format, we specify the format twice in the prompt—once in the middle and again at the end. For brevity, this repeated instruction (i.e., at the end) is omitted here. Please refer to our code repository for full details.

You are an expert machine learning researcher and your task is to evaluate a paper. You will be given a machine learning paper, which is based on a task description, a research idea, a literature review, a research proposal and experimental results. You will evaluate the paper on a scale of 1 to 10 across four key dimensions: Consistency, Clarity, Completeness, Soundness and finally give an overall assessment on a scale of 1 to 10. Please be objective in your evaluation, and provide detailed justifications for each score you assign. Do not hesitate to assign lower scores if the paper does not fully meet the criteria. Avoid giving high scores by default.

## Evaluation Rubric
{Writing Quality Evaluation Rubrics as shown in Table 27}
## Output Format

Please evaluate the paper according to the rubric and output a complete JSON object strictly following the format below, including all evaluation items (Consistency, Clarity, Completeness, Soundness, OverallAssessment). Do not output only a single item or partial content; you must output the entire JSON object.

```
{
    "Consistency": {
        "score": <1-10>,
        "justification": "<detailed explanation of why the score
            was given, referencing the consistency of the paper
            itself and between the paper and the task description,
            research idea, research proposal and experimental
            results>"
    },
    "Clarity": {
        "score": <1-10>,
        "justification": "<detailed explanation of why the score
            was given, referencing the clarity of the paper writing,
             structure, and coherence>"
    },
    "Completeness": {
        "score": <1-10>,
        "justification": "<detailed explanation of why the score
            was given, referencing the completeness of the paper in
            addressing the task description, research idea, research
             proposal and experimental results>"
    },
    "Soundness": {
        "score": <1-10>,
        "justification": "<detailed explanation of why the score
            was given, referencing the soundness of the paper's
            arguments, findings, methodology, and experimental
            results>"
    },
    "OverallAssessment": {
        "score": <1-10>,
        "strengths": ["<strength 1>", "<strength 2>"],
        "weaknesses": ["<weakness 1>", "<weakness 2>"]
    }
}
```

Please make sure the answer is strictly in valid JSON format.
- Do not include any text, comments, or explanations outside the JSON code block.
- Do not use trailing commas.
- Do not use single quotes; use double quotes for all keys and string values.
- Do not include comments inside the JSON.
- Do not use unescaped control characters (e.g., newlines inside strings).
- Do not use unquoted keys; all keys must be in double quotes.
- Do not use invalid values like NaN or Infinity.
- Ensure all brackets and braces are properly closed.

Table 32: Instruction prompt used for evaluating the quality of the experimentation process. The format is specified twice in the prompt—midway and at the end—to guide the model. We omit the final repetition here for brevity.

You are an expert machine learning researcher and your task is to evaluate a machine learning experimental document. You will be given a document containing experimental execution records and experimental results, which is based on a task description, a research idea, a literature review and a research proposal. You will first determine if the document contains any hallucinated content, then evaluate the document on a scale of 1 to 10 across six key dimensions: Consistency, Completeness, Novelty, Soundness, Insightfulness, Significance and finally give an overall assessment on a scale of 1 to 10. Please be objective in your evaluation, and provide detailed justifications for each score you assign. Do not hesitate to assign lower scores if the experimental document does not fully meet the criteria. Avoid giving high scores by default.

## Evaluation Rubric
{Experimentation Evaluation Rubrics as shown in Table 26}

## Output Format
Please evaluate the experimental document according to the rubric and output a complete JSON object strictly following the format below, including all evaluation items (Hallucination, Consistency, Completeness, Novelty, Soundness, Insightfulness, Significance, OverallAssessment). Do not output only a single item or partial content; you must output the entire JSON object.

```
{
    "Hallucination": {
        "has_hallucination": <true/false>,
        "details": "<if has_hallucination is true, provide specific
            examples of hallucinated content; if false, explain why
            you believe there is no hallucination>"
    },
    "Consistency": {
        "score": <1-10>,
        "justification": "<detailed explanation of why the score
            was given, referencing the alignment with the task
            description, idea, and literature review>"
    },
    "Completeness": {
        "score": <1-10>,
        "justification": "<detailed explanation of why the score
            was given, referencing the inclusion of necessary
            experiments, baselines, and ablation studies>"
    },
    "Novelty": {
        "score": <1-10>,
        "justification": "<detailed explanation of why the score
            was given, referencing the originality and innovation of
            the findings, methods and experimental design>"
    },
    "Soundness": {
        "score": <1-10>,
        "justification": "<detailed explanation of why the score
            was given, referencing the logical soundness and
            scientific rigor of the experimental design, analysis,
            and conclusions>"
    },
    "Insightfulness": {
        "score": <1-10>,
        "justification": "<detailed explanation of why the score
            was given, referencing the depth of insights, meaningful
            interpretations, and valuable implications for the
            field>"
    },
    "Significance": {
        "score": <1-10>,
        "justification": "<detailed explanation of why the score
            was given, referencing the importance and impact of the
            experimental results for the field>"
    },
    "OverallAssessment": {
        "score": <1-10>,
        "strengths": ["<strength 1>", "<strength 2>"],
        "weaknesses": ["<weakness 1>", "<weakness 2>"]
    }
}
```

Please make sure the answer is strictly in valid JSON format.

- Do not include any text, comments, or explanations outside the JSON code block.
- Do not use trailing commas.
- Do not use single quotes; use double quotes for all keys and string values.
- Do not include comments inside the JSON.
- Do not use unescaped control characters (e.g., newlines inside strings).
- Do not use unquoted keys; all keys must be in double quotes.
- Do not use invalid values like NaN or Infinity.
- Ensure all brackets and braces are properly closed.

Table 33: Instruction prompt used for evaluating the quality of the generated research proposal.

You are an expert machine learning researcher! You will be given a research proposal based on a task description, a research idea, and a literature review. Your task is to evaluate the proposal on a scale of 1 to 10 across six key dimensions and finally give an overall assessment on a scale of 1 to 10. Please be objective in your evaluation, and provide detailed justifications for each score you assign. Do not hesitate to assign lower scores if the proposal does not fully meet the criteria. Avoid giving high scores by default.

## Evaluation Rubric
{Research Proposal Rubrics as shown in 25}

## Output Format
Please output a complete JSON object strictly following the format below, including all evaluation items (Consistency, Clarity, Novelty, Soundness, Feasibility, Significance, OverallAssessment). Do not output only a single item or partial content; you must output the entire JSON object. When writing mathematical formulas, you should avoid invalid escape JSON decode errors.

```
{
    "Consistency": {
        "score": <1-10>,
        "justification": "<detailed explanation of why the score
            was given, referencing the alignment with the task
            description, idea, and literature review>",
    },
    "Clarity": {
        "score": <1-10>,
        "justification": "<detailed explanation of why the score
            was given, referencing the clarity of the proposal>",
    },
    "Novelty": {
        "score": <1-10>,
        "justification": "<detailed explanation of why the score
            was given, referencing the originality and innovation of
             the proposal>",
    },
    "Soundness": {
        "score": <1-10>,
        "justification": "<detailed explanation of why the score
            was given, referencing the technical foundations and
            rigor of the proposal>",
    },
    "Feasibility": {
        "score": <1-10>,
        "justification": "<detailed explanation of why the score
            was given, referencing the practicality and
            implementability of the proposal>",
    },
    "Significance": {
        "score": <1-10>,
        "justification": "<detailed explanation of why the score
            was given, referencing the importance and impact of the
            proposal>",
    },
    "OverallAssessment": {
        "score": <1-10>,
        "strengths": ["<strength 1>", "<strength 2>"],
        "weaknesses": ["<weakness 1>", "<weakness 2>"]
    }
}
```

Please make sure the answer is strictly in valid JSON format.
- Do not include any text, comments, or explanations outside the JSON code block.
- Do not use trailing commas.
- Do not use single quotes; use double quotes for all keys and string values.
- Do not include comments inside the JSON.
- Do not use unescaped control characters (e.g., newlines inside strings).
- Do not use unquoted keys; all keys must be in double quotes.
- Do not use invalid values like NaN or Infinity.
- Ensure all brackets and braces are properly closed.

# E   Human Study Details

To validate the effectiveness of **MLR-Judge**, we compare its evaluations against those of human reviewers. Specifically, we recruited 10 participants with prior reviewing experience at top-tier conferences aligned with our task domains, including ICML, NeurIPS, and ICLR. Based on their areas of expertise, each reviewer was assigned a subset of AI-generated papers relevant to their

domain. Each reviewer received the research paper along with its corresponding supplementary code. We collected all responses through Google Forms.

The evaluation criteria followed the same rubric used in the end-to-end setting of MLR-Judge (see Table 28). Fig. 8 and Fig. 9 show the Google Form interface used for receiving the human review data. All collected human review results are available as a CSV file in our GitHub repository under the `human-study` folder.

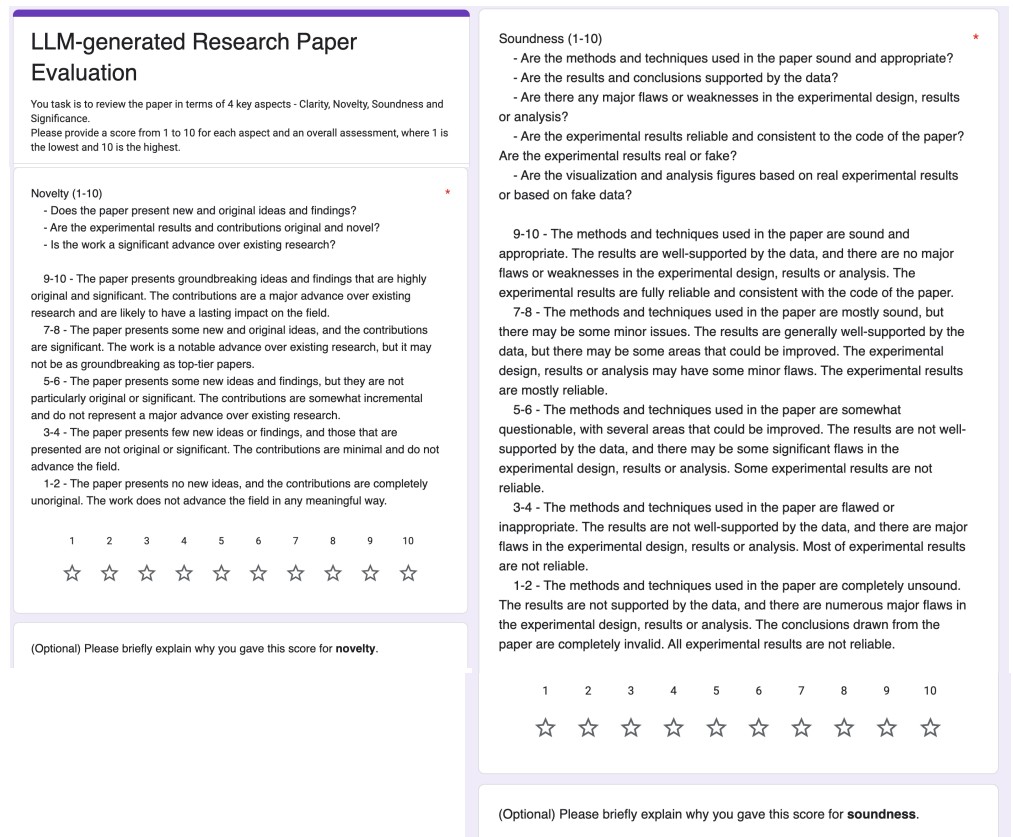

Figure 8: The Google Form interface used to collect review scores, following the same reviewer rubrics as MLR-Judge (Table 28).

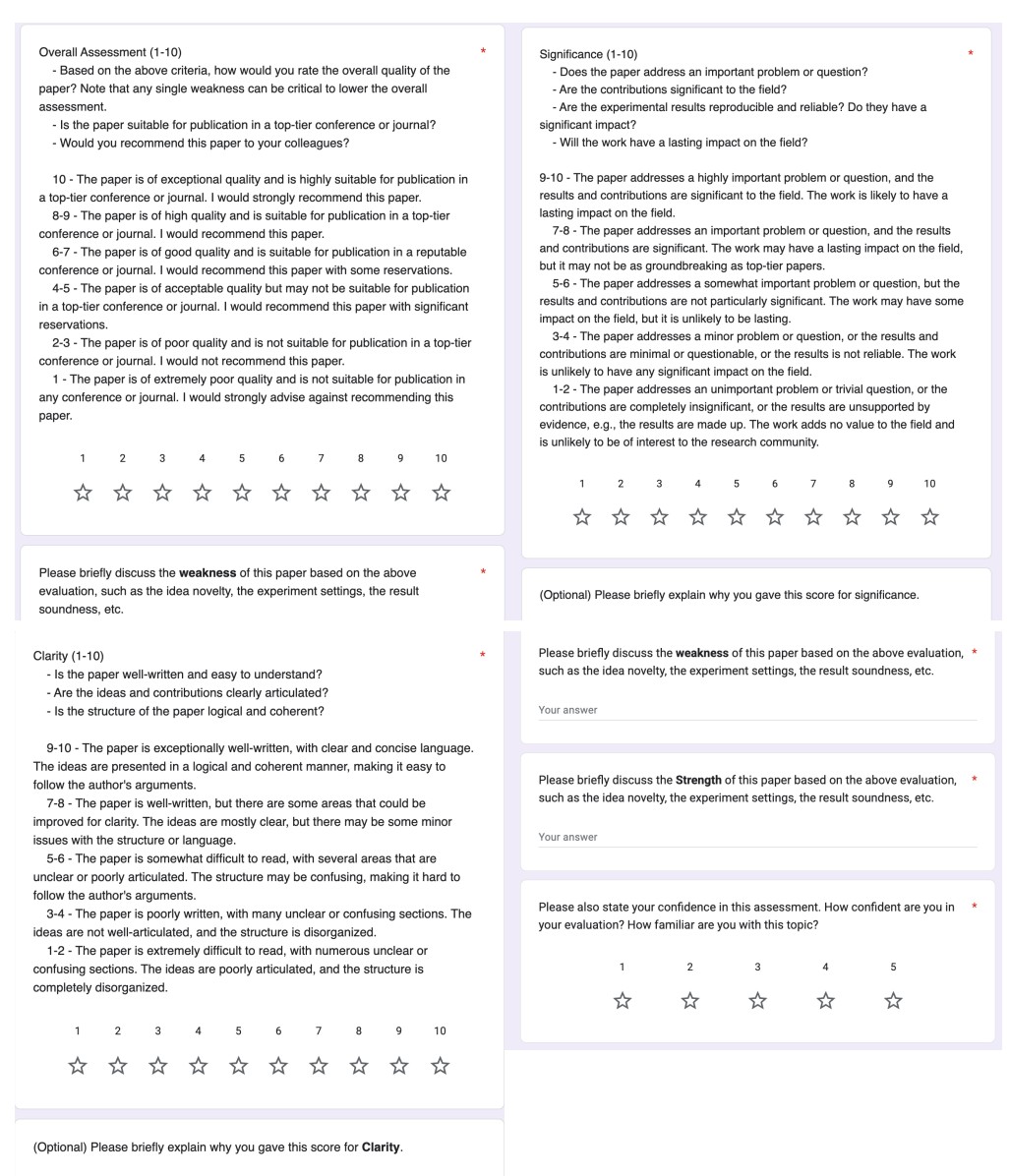

Figure 9: The Google Form interface used to collect review scores, following the same reviewer rubrics as MLR-Judge (Table 28).

