# OpenReview forum: "MLR-Bench: Evaluating AI Agents on Open-Ended Machine Learning Research"
_NeurIPS.cc/2025/Datasets_and_Benchmarks_Track — NeurIPS 2025 Datasets and Benchmarks Track poster_

### Official Review · Reviewer_ESS7 · 2025-06-01

**Rating:** 4
**Confidence:** 4

**Summary:**

The paper introduces MLR-Bench, a comprehensive benchmark to evaluate AI agents on open-ended machine learning research tasks spanning ideation, proposal writing, experimentation, and paper composition. It features a modular agent (MLR-Agent) and an automated LLM-based reviewer (MLR-Judge), validated against expert human reviewers. Through extensive evaluation on 201 tasks, the study highlights current LLMs' strengths in generating coherent ideas and papers, but also reveals major weaknesses—particularly in experimental reliability and novelty.

**Dataset Code Accessibility:**

Partly

**Ethical Considerations:**

No, there are no or only very minor ethics concerns

**Final Justification:**

The paper makes valuable contributions to an important problem space and demonstrates thorough empirical work within its constraints. However, the methodological limitations—particularly around evaluation circularity, reproducibility, and experimental scope—prevent this from being a robust, reliable benchmark that the community can build upon with confidence. A score of 4 reflects work with merit but fundamental limitations that restrict its impact.

**Limitations Weaknesses:**

1. Evaluation Relies Exclusively on LLM-Based Judges

While MLR-Judge is carefully designed and validated against human reviewers (Section 4), the entire evaluation pipeline ultimately depends on AI models scoring other AI models, introducing potential circularity. This is particularly critical in a benchmark context, where reproducibility and grounding in objective criteria are paramount. The LLM judges may implicitly favor fluency or stylistic quality over actual correctness or methodological rigor, especially in dimensions like Soundness or Feasibility. Figure 4 confirms this concern to some extent — the p-value for Clarity (~0.055) hints at a nontrivial misalignment with human judgments.

2. Human Evaluation Underdocumented

Although Section 4 mentions that 10 NeurIPS/ICLR-experienced reviewers evaluated model outputs, the paper does not report how many total papers were human-reviewed, nor how many examples each reviewer saw. This lack of transparency makes it difficult to assess the robustness and generalizability of the human-LLM agreement results (e.g., those presented in Table 5 and Figure 4), which are central to justifying the validity of MLR-Judge as a benchmark component.

3. Ambiguity and Subjectivity in Evaluation Rubrics

The benchmark evaluates outputs on multi-dimensional rubrics (e.g., Novelty, Clarity, Feasibility, Soundness, Insightfulness), but some of these criteria are qualitative and loosely defined, especially in edge cases or across task stages. For instance, “Clarity” is used both in idea generation and paper writing but likely captures different aspects in each. The rubric definitions (Table 1) provide high-level guidance, but no scoring anchors or examples are given to clarify borderline scores. Additionally, the lack of inter-rater calibration between the two LLM judges increases the risk of scoring variance.

4. Limited Experimental Evaluation on Downstream Tasks

While the benchmark includes 201 tasks in total, only 10 tasks are used in the coding and paper-writing stages (Sections 3.3–3.4), which means downstream performance metrics are based on a narrow slice of the benchmark. This is especially concerning for the experimentation stage, where one of the key findings (i.e., that 80% of experiments contained hallucinated results) is based on a very limited set. For a benchmark paper, the downstream evaluation should ideally reflect the diversity of the full task set.

5. Incomplete Coverage of Agent Variants and Ablations

The MLR-Agent framework (Section 2.2) is evaluated with a small set of models and configurations, but the paper does not explore agent-level ablations or modular design comparisons (e.g., the impact of different planning strategies, tool use, or memory components). For a benchmark paper introducing a reference agent, more thorough evaluation of architectural alternatives would strengthen its value as a baseline.

6. Use of Proprietary and Preview-Only Models

Several of the evaluated models (e.g., Gemini-2.5-Pro-Preview, Claude-3.7-Sonnet, o4-mini) are not publicly available or are in restricted preview. This significantly limits replicability and comparability, particularly for researchers aiming to reproduce or extend the benchmark. Table 2 does not provide enough detail on exact versions or inference settings, which hinders transparency — a critical requirement for any benchmark claiming to set new evaluation standards.

7. No Ground-Truth Verification Possible

The open-ended research tasks intentionally avoid ground-truth answers, but this creates a limitation: output quality is assessed purely by perceived plausibility and coherence, not correctness. In a benchmark setting, this undermines the possibility of objective regression testing or stable progress tracking across iterations of models or agents. Without any reference outputs or canonical solutions, score interpretability is also limited.

8. Observed Shortcut Behavior in Execution Agent

The Claude-based execution agent repeatedly fabricated experimental results when code execution failed (Section 3.3, Figure 5), despite being instructed not to do so. This shortcut behavior is not unique to this agent and may appear across many LLM-based systems, but the benchmark lacks instrumentation or detection mechanisms to catch this behavior automatically. As a result, systems that generate plausible-sounding outputs despite internal failures may appear artificially strong under this benchmark.

**Strengths Contributions:**

The paper’s strengths lie in its innovative benchmark design, the thorough evaluation with multiple cutting-edge models, and the insightful analysis of where AI agents succeed or fail in a research context. The work is timely and addresses an important challenge for the future of AI in science. The fact that the evaluation framework (MLR-Judge) was validated against human experts adds a lot of credibility. The findings (like the prevalence of fabricated experiment results) are not only interesting but also actionable, pointing researchers toward specific problems to solve.

---

> ### Author Rebuttal · Authors · 2025-07-31
>
> We sincerely thank the reviewer for recognizing the novelty of our benchmark design, the thoroughness of our evaluation, and the insightfulness of our analysis. We especially appreciate the acknowledgment of our efforts to validate MLR-Judge against human experts, and we are encouraged that findings—such as the prevalence of fabricated experimental results—are viewed as both meaningful and actionable. Below, we address all the questions raised and hope our responses effectively resolve the concerns.
>
> ---
>
> ### **W1: Evaluation Relies Exclusively on LLM-Based Judges**
>
> Thank you for this thoughtful concern. This issue has been a very important consideration during the benchmark design phase, and we have adopted the following actions to address this:
>
> * **Human alignment study confirms validity.**
> While human evaluation is prohibitively costly at scale, we conducted a human alignment study with 10 expert reviewers (Section 4) to verify that our LLM-based evaluation remains reliable. The results show that **MLR-Judge’s scores are statistically indistinguishable from human ratings**. The distributions largely overlap, indicating that MLR-Judge can approximate human judgment reliably, especially at scale.
>
> * **Avoiding circularity through model separation**
>  To mitigate the risk of circular evaluation (e.g., using Claude to judge outputs from Claude), we employ two independent LLM judges—Gemini and Claude—and ensure that no model is evaluated solely by its own backbone family. This separation helps prevent self-reinforcing bias.
>
> * **Rubric-guided prompting to reduce fluency bias**
> To avoid the risk that LLM judges may favor fluency over rigor, we carefully designed the review rubrics for each state that explicitly guide the evaluation towards technical quality, factual grounding, and methodological rigor. For example, Table 25 (Appendix B.2.1) shows how writing outputs are judged based on criteria like Soundness and Completeness rather than just linguistic quality.
>
> * **Dedicated hallucination detection**
> In the experimentation stage, we introduce a separate hallucination detection metric (Supplementary B.2.2) that targets fabricated or ungrounded content. This is supported by targeted prompts that ask the judge to verify whether claimed results are supported by code execution logs and experiment history. By explicitly evaluating factual validity, we ensure that Soundness is assessed beyond surface-level fluency.
>
> We fully acknowledge that our LLM-based evaluation is not perfect, as pointed out by the reviewer. Our goal is to **offer a scalable and reasonably accurate alternative to human reviews**, and we hope our design serves as a solid foundation toward that end. We also view this as an **open area for future research**, and encourage follow-up work to explore better LLM-judges.
>
> ---
>
> ### **W2: Human Evaluation Underdocumented**
>
> Thank you for pointing this out. We provide the following clarifications:
>
> * We recruited **10 reviewers with experience at NeurIPS/ICLR**.
> * Each reviewer was assigned **a subset AI-generated papers** relevant to their expertise.
> * Each paper was evaluated by 2 reviewers using the **OpenReview-style rubric** (see Appendix C and Figure 6/7 Google Doc Interface), covering Novelty, Soundness, Clarity, Significance, and Overall Score.
> * For transparency, **individual scores and reviewer comments** are included in our **GitHub repo human_eval folder**.
>
> ---
>
> ### **W3: Ambiguity and Subjectivity in Evaluation Rubrics**
>
> Thank you for raising this important point. The exact definition and prompt template can be seen in Appendix B of the supplementary. We address the concern below:
>
> * **Stage-specific rubric definitions**
>   Although some criteria (e.g., *Clarity*) appear across multiple stages, we provide **stage-specific definitions** and prompt templates tailored to each context.
>   For example:
>   * *Clarity* in the **idea generation stage** refers to whether the research idea is *conceptually well-articulated and clearly framed*.
>   * *Clarity* in the **paper writing stage** refers to the *narrative coherence and readability* of the full manuscript.
>     These distinctions are encoded in the corresponding prompts and scoring descriptions (see **Appendix B**, Tables **21** and **25**).
> * **Rubric-guided prompting and structured inputs**
>   To reduce ambiguity, each LLM judge is guided by a **rubric-aligned prompt** that explicitly outlines what to evaluate and how to score it. These prompts include descriptive anchors (e.g., “7–8 = Good: mostly clear with only minor ambiguity”), as shown in Appendix B.2.2.
>
> * **Addressing inter-rater consistency**
>   While we acknowledge that our evaluation currently uses two separate LLM judges (Gemini and Claude) without explicit inter-rater calibration, we mitigate variance by:
>
>   * Averaging their scores for robustness.
>   * Using clearly defined rubric criteria.
>   * Conducting a **human alignment study** (Section 4), where we show that the variance across human reviewers is comparable to that of our LLM judges.
>
> We agree that providing more concrete borderline examples would further improve transparency, and we plan to include such scoring examples in future releases of MLR-Bench to aid reproducibility and interpretability.
>
> ---
>
> ### **W4: Limited Experimental Evaluation on Downstream Tasks**
>
> The decision to evaluate only 10 tasks in these stages was primarily due to cost and runtime constraints. Running full pipelines—including automated coding, experiment execution, and paper writing—is computationally intensive and time-consuming, particularly when repeated across multiple models.
>
> To balance coverage and feasibility, we carefully selected 10 tasks spanning diverse domains such as (multimodal) LLMs, trustworthy AI, and ML theory. This diversity was intended to ensure that the evaluation captures representative behaviors across different research areas and data modalities.
>
> Despite the limited scope, this subset already revealed consistent failure patterns, including shortcut-taking behaviors (e.g., hallucinated results, fabricated plots) and shortcomings in scientific soundness. These issues were observed across multiple models and evaluation stages, suggesting they are not task-specific anomalies but indicative of broader limitations in current AI research agents.
>
> We agree that scaling the evaluation to more tasks would further strengthen the generalizability of our findings. This remains a direction for future work. We hope the release of MLR-Bench will also facilitate broader community contributions and extensions to the benchmark.
>
>
> ---
>
> ### **W5: Incomplete Coverage of Agent Variants and Ablations**
>
> Thank you for the suggestion. We note:
>
> * MLR-Bench is designed to support **arbitrary research agents**. Our goal is to build an **agent-agnostic evaluation framework**.
> * We have already evaluated **AI-Scientist v2** using MLR-Judge, and observed consistent trends (Appendix A.2).
> * Modular ablations (e.g., planning vs. non-planning) are **not explored in this version**, but we see this as an exciting future direction.
> * We welcome future work to benchmark additional **agent variants** under a unified setting.
>
> ---
>
> ### **W6: Use of Proprietary and Preview-Only Models**
>
> Thank you for your suggestion! For the judge model, we use `claude-3-7-sonnet-20250219` and `gemini-2.5-pro-preview-05-06`.
> At the time of our experiments, these proprietary models were among the few capable of handling multi-modal reasoning, a critical requirement for end-to-end evaluation of research workflows. Tasks such as interpreting plots, experiment figures, and visual outputs demand robust vision-language capabilities. Unfortunately, most open-source models available at the time (e.g., DeepSeek-R1, Ministral-8B) lacked image input support and thus were not viable for this setting.
>
> We fully support the use of open-source models and encourage future work to re-run MLR-Bench using publicly available agents as vision and reasoning capabilities continue to improve in the open-source community. MLR-Bench is designed to be model-agnostic and extensible, allowing for ongoing benchmarking across evolving systems.
>
> ---
>
> ### **W7: No Ground-Truth Verification Possible**
>
> Indeed, open-ended research tasks **do not have fixed ground truths**. However:
>
> * We follow **human review-style rubrics** for evaluation, similar to real conference reviews (see Appendix B.2).
> * For factual verification, we introduce **hallucination detection metrics** in the experimentation stage.
> * The goal is not to enforce one “correct answer,” but to **encourage insight, rigor, and creativity**—key traits in real-world research.
>
> ---
>
> ### **W8: Observed Shortcut Behavior in Execution Agent**
>
> Thank you for pointing this out. We observed **Claude’s coding agent fabricating experimental results** when execution failed, despite being prompted otherwise (Figure 5).
>
> * This behavior is **not unique** to Claude—many LLMs tend to hallucinate when unsure.
> * MLR-Bench includes **manual hallucination checks** in Section 3.3, but does **not yet include automated detectors**.
> * We agree that future benchmarks should include **instrumentation for failure detection**, e.g., execution tracing, return code monitoring, or hallucination classifiers.

---

> > ### Author Response · Authors · 2025-08-01
> > **Comments by Authors**
> >
> > To supplement **W5**, we would add more evaluation results of **MLR-Agent** and **AI Scientist V2 [1]** in Table B. Both agents are powered by `o4-mini-high`. Since **Codex** is now capable of executing experiments fully autonomously with network access, we use **Codex** as the coding agent in MLR-Agent.
> >
> > 1. *Table B: End-to-end performance comparison of AI Scientist V2 and MLR-Agent on the ten tasks, based on the average ratings of two judges. MLR-Agent uses Codex as the coding agent. As Codex only supports a `reasoning_effort of medium` now, we use `o4-mini-medium` as its engine.*
> > | Agent Scaffold | Clarity | Novelty | Soundness | Significance | Overall |
> > | :--- | :--- | :--- | :--- | :--- | :--- |
> > | MLR-Agent | 6.45 ± 0.90 | 5.65 ± 0.60 | 2.90 ± 0.57 | 3.80 ± 0.65 | 3.10 ± 0.60 |
> > | AI Scientist V2 | 6.55 ± 0.94 | 6.70 ± 0.48 | 3.70 ± 1.29 | 4.85 ± 1.08 | 4.25 ± 1.25 |
> >
> > As shown in Table B, **AI Scientist V2** outperforms **MLR-Agent** across all evaluation dimensions. It is noticeable that both agents share similar weaknesses—struggling with *Soundness* and *Significance*—highlighting crucial limitations in current research agents.
> >
> > In addition, we would include end-to-end evaluation results for OpenAI **Codex** and Google **Gemini CLI**, which are now capable of executing experiments fully autonomously with network access. Since **Codex** currently only supports a `reasoning_effort of medium`, we use `o4-mini-medium` as its engine.
> >
> > 2. *Table A: End-to-end evaluation results over the ten tasks. We report the average ratings of two judges. We use `gemini-2.5-pro-preview-05-06` as the engine of Gemini-2.5-Pro-Preview  + Gemini CLI.*
> > | Model | Clarity | Novelty | Soundness | Significance | Overall |
> > | :--- | :--- | :--- | :--- | :--- | :--- |
> > | o4-mini-high + Codex | 6.45 ± 0.90 | 5.65 ± 0.60 | 2.90 ± 0.57 | 3.80 ± 0.65 | 3.10 ± 0.60 |
> > | o4-mini-high + Claude Code | 7.65 ± 0.32 | 6.90 ± 0.30 | 3.80 ± 0.60 | 5.05 ± 0.57 | 3.95 ± 0.71 |
> > | Gemini-2.5-Pro-Preview  + Gemini CLI | 8.30 ± 0.37 | 6.85 ± 0.34 | 4.15 ± 1.06 | 5.30 ± 0.91 | 4.60 ± 1.00  |
> > | Claude-3.7-Sonnet + Claude Code | 7.75 ± 0.34 | 7.10 ± 0.43 | 4.05 ± 1.11 | 5.50 ± 1.14 | 4.70 ± 1.22  |
> >
> > This table shows that, **Claude Code** outperforms **Codex** and **Gemini CLI** in end-to-end research, and `Claude-3.7-Sonnet` works better than `o4-mini-high`.
> >
> > [1] The AI Scientist-v2: Workshop-Level Automated Scientific Discovery via Agentic Tree Search.

---

> > > ### Comment · Reviewer_ESS7 · 2025-08-06
> > >
> > > The paper makes contributions (comprehensive task collection, attempt at automated evaluation) but has significant methodological weaknesses that limit its reliability as a benchmark. The rebuttal adds details but doesn't address the core architectural problems. A score of 4 appropriately reflects a paper with interesting ideas but fundamental limitations that prevent stronger endorsement.
> > > The authors' transparency about limitations is appreciated but doesn't fix them

---

> > > > ### Author Response · Authors · 2025-08-07
> > > >
> > > > We sincerely thank Reviewer ESS7 for the feedback. In our rebuttal, we have provided detailed descriptions to explain the strategies we have tried in this work to address the methodological weaknesses that Reviewer ESS7 pointed out.
> > > >
> > > > For instance, regarding the question of *why MLR-Judge relies exclusively on LLM-based judges*: MLR-Judge aims to **mimic human evaluation** in paper review. While not a perfect solution, LLM judges are **currently the most intelligent and scalable approach for large-scale assessment**. Furthermore, to mitigate potential bias, we have **averaged the outputs from multiple LLM judges**.
> > > >
> > > > Furthermore, we believe our paper's primary contributions are well-aligned with the scope of the DB track:
> > > >
> > > > (1) We have carefully collected and released a **new open-source dataset** containing 201 tasks derived from previously available data.
> > > >
> > > > (2) We have proposed MLR-Bench, a **new benchmark** that includes our MLR-Judge as a **benchmark tool**, to evaluate AI agents on open-ended autonomous machine learning research.
> > > >
> > > > (3) Moreover, we have conducted systematic and in-depth analyses of the challenges for current research agents, which provides important new insights into this area.
> > > >
> > > > Finally, we would appreciate it if Reviewer ESS7 could provide more specific details regarding the unresolved methodological weaknesses. This would help us to better understand and address the concerns.

---

### Official Review · Reviewer_xPbu · 2025-06-10

**Rating:** 5
**Confidence:** 4

**Summary:**

This paper introduces MLR-Bench, a new benchmark designed to evaluate the capabilities of AI agents in conducting open-ended machine learning research. The benchmark is composed of three main parts: (1) a set of research tasks derived from top-tier ML conference workshops; (2) MLR-Judge, an automated evaluation framework that uses large language models (LLMs) with structured rubrics to assess research quality ; and (3) MLR-Agent, a modular agent designed to perform research tasks through stages of idea generation, proposal, experimentation, and paper writing. The authors use this framework to evaluate several state-of-the-art LLMs and a coding agent. A key finding is that while LLMs are proficient at ideation and writing, current agents frequently fail in the experimentation phase, often producing fabricated results to mask execution failures. The paper also presents a human evaluation study that shows a high level of agreement between MLR-Judge and expert human reviewers, suggesting its viability as a scalable evaluation tool

**Additional Feedback:**

NA

**Dataset Code Accessibility:**

Yes

**Dataset Code Comments:**

https://huggingface.co/datasets/chchenhui/mlrbench-tasks

**Ethical Considerations:**

No, there are no or only very minor ethics concerns

**Final Justification:**

>> After rebuttal
raise my score

I think it would be valuable to the community.

**Limitations Weaknesses:**

1. The experimentation phase evaluation in Section 3.3 relies solely on a single coding agent. While the findings are striking, attributing the "shortcut-taking behavior"  to the broader class of AI research agents might be an overstatement based on an N of 1. The conclusions about experimental reliability would be much stronger if they were shown to hold across multiple, different coding agents
2. The experimentation, paper writing, and end-to-end evaluations (Sections 3.3, 3.4, 3.5) are conducted on a manually selected subset of only 10 tasks

**Strengths Contributions:**

1.  The problem of evaluating and improving AI agents for scientific discovery is of great importance. This paper tackles the challenge head-on by providing a comprehensive, end-to-end benchmark.
2. MLR-Bench is thoughtfully constructed. I particularly like the decomposition of the research process into four distinct stages, which allows for both "stepwise" and "end-to-end" evaluation.

---

> ### Author Rebuttal · Authors · 2025-07-31
>
> **W1. The experimentation phase evaluation in Section 3.3 relies solely on a single coding agent. While the findings are striking, attributing the "shortcut-taking behavior" to the broader class of AI research agents might be an overstatement based on an N of 1. The conclusions about experimental reliability would be much stronger if they were shown to hold across multiple, different coding agents**
>
> Thank you for this insightful comment. We agree that broader coverage of coding agents would strengthen the generality of our findings. In this work, we chose Claude Code primarily because, at the time of submission (May 16, 2025), it was the only available coding agent that supported fully autonomous code execution with internet access. At that time, other coding agents like Codex only allowed fully autonomous code execution in a sandbox without network access.
>
> We have to clarify that our current analysis and conclusions are based solely on Claude Code. With different coding agents emerging in the future, our conclusions will evolve. We also agree that, to draw a more generalized conclusion, it would indeed be necessary to wait for multiple coding agents to become available. This does not diminish the contribution of our benchmark; MLR-Bench is designed to help researchers analyze the performance of coding agents in automated machine learning research.
>
> ----
>
> **W2. The experimentation, paper writing, and end-to-end evaluations (Sections 3.3, 3.4, 3.5) are conducted on a manually selected subset of only 10 tasks**
>
> For the end-to-end evaluation stages, we carefully selected a diverse set of 10 tasks to **balance evaluation depth with resource feasibility**. This approach was necessitated by the significant **computational cost and runtime** required for full pipeline execution across multiple models.
> Moreover, our subset spans diverse domains such as (multimodal) LLMs, trustworthy AI, and ML theory. This diversity was intended to ensure that the evaluation captures representative behaviors across different research areas and data modalities.
>
> While we agree that scaling the evaluation would further strengthen generalizability—a direction we leave for future work—we argue that our core contribution is already demonstrated. We have to mention that, beyond evaluation, our MLR-Bench is able to provide valuable **feedback signals** for **improving research agent training**. Our evaluation results on the current subset already revealed consistent failure patterns, including shortcut-taking behaviors (e.g., hallucinated results, fabricated plots) and shortcomings in scientific soundness. These issues, observed across multiple models and stages, pinpoint key limiting factors in current research agent training.

---

> > ### Author Response · Authors · 2025-08-01
> > **Comments by Authors**
> >
> > To supplement **W1**, we have added evaluation results of **Codex** in experiment execution and compared its performance with that of **Claude Code**. The results are presented in Table C. **Codex** is now capable of executing experiments fully autonomously with network access. Since Codex currently only supports a `reasoning_effort of medium`, we use `o4-mini-medium` as its engine.
> >
> > *Table C: Evaluation results of Claude Code and Codex averaged on ten tasks in experiment execution, based on the average ratings of two judges.*
> > | Coding Agent | Consistency | Completeness |  Novelty | Soundness | Insightfulness | Significance | Overall |
> > | :--- | :--- | :--- | :--- | :--- | :--- | :--- | :--- |
> > | Claude Code | 6.75 ± 1.00 | 6.00 ± 0.68 | 5.65 ± 0.82 | 4.75 ± 1.02 | 4.50 ± 0.97 | 4.70 ± 0.95 | 4.95 ± 0.82 |
> > | Codex | 6.30 ± 1.46 | 5.05 ± 0.87 | 3.80 ± 1.19 | 6.15 ± 0.87 | 4.45 ± 1.28 | 3.40 ± 1.05 | 4.95 ± 1.02 |
> >
> > Table C revealed weaknesses in **Claude Code** regarding *Soundness*, *Insightfulness*, and *Significance*, and its *Overall* score was below the 6.0 acceptance threshold. These findings have already been summarized in our current submission. Additionally, we observed that **Codex** obtained low ratings in *Completeness*, *Novelty*, *Insightfulness*, and *Significance*, while it received ratings over 6.0 in *Consistency* and *Soundness*.
> >
> > We analyzed the reviews and came to the following conclusions:
> > 1. The experimental results of **Codex** are highly reproducible, and thus its *Soundness* score is higher than 6.0.
> > 2. The experimental process of **Codex** was weak, leading to a low rating in *Completeness*. For instance, it often used partial datasets, shortened training procedures, and omitted necessary ablation studies.
> > 3. The model framework implemented by **Codex** was less novel than that of **Claude Code**, accounting for its lower score in *Novelty*.
> > 4. Experimental results of both *Claude Code* and *Codex* lack *Significance* in automated machine learning research.
> >
> > We hope our new results and findings will give more insights on coding agents in automated research.

---

> > > ### Comment · Reviewer_xPbu · 2025-08-04
> > >
> > > Thanks for the reply. I think it is an interesting work. Raised my score.

---

> > > > ### Author Response · Authors · 2025-08-04
> > > >
> > > > Thank you for reviewing our response. We appreciate your positive feedback and the updated score!

---

### Official Review · Reviewer_9DJw · 2025-06-20

**Rating:** 5
**Confidence:** 4

**Summary:**

This paper introduce MLR-Bench, a benchmark for evaluating AI agents on open-ended machine learning research. MLR-Bench includes (1) 201 research tasks; (2) MLR-Judge, an automated evaluation framework combining LLMbased reviewers; (3) MLR-Agent, a modular agent scaffold capable of completing research tasks through four stages: idea generation, proposal formulation, experimentation, and paper writing. The authors use MLR-Bench to evaluate six frontier LLMs and an advanced coding agent and validate MLR-Judge through human evaluation, showing high agreement with expert reviewers, supporting its potential as a scalable tool for research evaluation.

**Additional Feedback:**

Please discuss my questions in the Weakness part. Thank you.

**Dataset Code Accessibility:**

Yes

**Ethical Considerations:**

No, there are no or only very minor ethics concerns

**Final Justification:**

The authors provide sufficient discussions to solve my concerns. I will raise my score to 5.

**Limitations Weaknesses:**

1. This paper is limited to the domain of Artificial Intelligence. Could the proposed framework be generalized in other domains?
2. The evaluation of a given LLM with MLR-Judge depends on using this LLM as the core in MLR-Agent. Would the specific design of MLR-Agent impact the performance of different LLMs? Is it possible to add some automated pipelines other than MLR-Agent, like AI-Researcher [1], AI-Scientist [2]?
3. Its not very clear how the pipeline and prompts of MLR-Judge is designed.

[1] Can llms generate novel research ideas? a large-scale human study with 100+ nlp researchers.
[2] The ai scientist: Towards fully automated open-ended scientific discovery.

**Strengths Contributions:**

1. This paper consider Novelty, Feasibility, etc. in their evaluation. These aspects are important for a good research idea.
2. This paper includes human-in-the-loop evaluation, showing that the obtained scores of ideas agree with human experts’ perspectives.
3. This paper provides some insightful discussions regarding the key factors affecting AI-generated research quality.

---

> ### Author Rebuttal · Authors · 2025-07-31
>
> We thank reviewer 9DJw for recognizing the importance of our evaluation aspects and for appreciating that our evaluation pipeline aligns with human expert judgments. We are glad that the reviewer finds our discussions insightful. Below, we address all the points raised and hope our responses adequately resolve the concerns.
>
> ---
>
> **Domain Generalization: This paper is limited to the domain of Artificial Intelligence. Could the proposed framework be generalized in other domains?**
>
> Thank you for the thoughtful question!
>
> - **Theoretical generalizability**:
>   The framework is **not AI-specific** in its design. The overall workflow—task definition, stepwise research generation, and LLM-based evaluation—can, in principle, apply to other scientific domains.
>
> - **Practical limitations**:
>   We also want to point out that there might be practical challenges when extending to other domains. Certain domains may lack open-access datasets or papers, and some experimental procedures may be **non-automatable** (e.g., those requiring physical setups or domain-specific tools).
>
> - **Promising directions**:
>   Fields like **mathematics** or **theoretical physics**, where research is mostly symbolic or text-based, are particularly well-suited for future extension.
>
> ----
>
> **Modularity of MLR-Agent: The evaluation of a given LLM with MLR-Judge depends on using this LLM as the core in MLR-Agent. Would the specific design of MLR-Agent impact the performance of different LLMs? Is it possible to add some automated pipelines other than MLR-Agent, like AI-Researcher [1], AI-Scientist [2]?**
>
> Thank you for raising this point!
>
> - **Not tied to a specific agent**:
>   We would like to point out that our proposed MLR-Bench framework is not tied to this specific agent design. One of our key goals is to build a general-purpose evaluation pipeline that can support a wide range of AI research agents, including external systems like AI-Scientist and AI-Researcher.
>
> - **Demonstrated extensibility**:
>   We also evaluated **AI-Scientist v2** using MLR-Bench (see Appendix A.2). Under the same backbone model (*o4-mini*), both AI-Scientist and MLR-Agent show similar behavior: achieve relatively higher scores in Novelty and Clarity but underperform in Soundness and Significance.
>
> We encourage researchers to use our MLR-Bench to evaluate their research agents and to foster **standardized, comparative evaluation**.
>
> ---
>
> **Design of MLR-Judge: It's not very clear how the pipeline and prompts are constructed.**
>
> Thank you for the feedback! We’re happy to clarify:
>
> - **Stage-wise decomposition**:
> Our goal was to build an evaluation framework that mirrors how human reviewers assess research at different stages of the scientific process. To this end, we decompose the research workflow into **four stages: idea generation, proposal writing, experimentation, and paper writing**.
>
> - **Rubrics design**:
> For each stage, we design evaluation prompts centered around criteria commonly used in academic peer review—such as those in NeurIPS and ICLR. For instance, Novelty is emphasized in the idea generation stage, while Soundness and Completeness are prioritized during experimentation.
>
> - **Prompt Input**
> Each prompt is constructed by combining the agent’s output with relevant contextual information from earlier stages (e.g., task description, generated idea, or proposal), allowing the LLM-as-judge to make context-aware and coherent assessments.
>
> - **Details can be found in Appendix B**:
>   - Review rubrics for each stage: **B.2.1**
>   - Prompt construction: **B.2.2**
>   - An example prompt (e.g., writing stage): **Table 25**

---

> > ### Author Response · Authors · 2025-08-01
> > **Comments by Authors**
> >
> > To supplement **W2: Would the specific design of MLR-Agent impact the performance of different LLMs?**, we have included end-to-end performance comparison of **AI Scientist V2** and **MLR-Agent** with `o4-mini-high` as the engine in Table B. **MLR-Agent** uses **Codex** as its coding agent. Since **Codex** currently only supports a `reasoning_effort of medium`, we use `o4-mini-medium` as its engine.
> >
> > *Table B: End-to-end performance comparison of AI Scientist V2 and MLR-Agent on the ten tasks, based on the average ratings of two judges.*
> > | Agent Scaffold | Clarity | Novelty | Soundness | Significance | Overall |
> > | :--- | :--- | :--- | :--- | :--- | :--- |
> > | MLR-Agent | 6.45 ± 0.90 | 5.65 ± 0.60 | 2.90 ± 0.57 | 3.80 ± 0.65 | 3.10 ± 0.60 |
> > | AI Scientist V2 | 6.55 ± 0.94 | 6.70 ± 0.48 | 3.70 ± 1.29 | 4.85 ± 1.08 | 4.25 ± 1.25 |
> >
> > This table shows that, using the same `o4-mini` model, **AI Scientist V2** outperforms **MLR-Agent**.
> >
> > Note that we did not add results of **AI Researcher** because it specializes in idea generation, whereas our benchmark requires the ability to do end-to-end research including conducting experiments.

---

> > > ### Author Response · Authors · 2025-08-04
> > >
> > > To supplement **W2**, we would also include end-to-end evaluation results for OpenAI **Codex** and Google **Gemini CLI**, which are now capable of executing experiments fully autonomously with network access. Since **Codex** currently only supports a `reasoning_effort of medium`, we use `o4-mini-medium` as its engine.
> > >
> > >  *Table A: End-to-end evaluation results over the ten tasks. We report the average ratings of two judges. We use `gemini-2.5-pro-preview-05-06` as the engine of Gemini-2.5-Pro-Preview  + Gemini CLI.*
> > > | Model | Clarity | Novelty | Soundness | Significance | Overall |
> > > | :--- | :--- | :--- | :--- | :--- | :--- |
> > > | o4-mini-high + Codex | 6.45 ± 0.90 | 5.65 ± 0.60 | 2.90 ± 0.57 | 3.80 ± 0.65 | 3.10 ± 0.60 |
> > > | o4-mini-high + Claude Code | 7.65 ± 0.32 | 6.90 ± 0.30 | 3.80 ± 0.60 | 5.05 ± 0.57 | 3.95 ± 0.71 |
> > > | Gemini-2.5-Pro-Preview  + Gemini CLI | 8.30 ± 0.37 | 6.85 ± 0.34 | 4.15 ± 1.06 | 5.30 ± 0.91 | 4.60 ± 1.00  |
> > > | Claude-3.7-Sonnet + Claude Code | 7.75 ± 0.34 | 7.10 ± 0.43 | 4.05 ± 1.11 | 5.50 ± 1.14 | 4.70 ± 1.22  |
> > >
> > > This table shows that, **Claude Code** outperforms **Codex** and **Gemini CLI** in end-to-end research, and `Claude-3.7-Sonnet` works better than `o4-mini-high`.

---

> > > > ### Comment · Reviewer_9DJw · 2025-08-05
> > > >
> > > > Thanks for your responses and additional results. The response has solved my major concerns. I believe this benchmark has potential benefit to the AI Scientist research community. I am going to raise my score to 5 in support of its acceptance.
> > > >
> > > > I am sorry for the late reply. Wish you good luck in future research.

---

> > > > > ### Author Response · Authors · 2025-08-05
> > > > >
> > > > > Thank you for your feedback and the updated score. We are very pleased that our responses addressed your concerns, and we appreciate your support!

---

### Official Review · Reviewer_k5SZ · 2025-07-03

**Rating:** 5
**Confidence:** 4

**Summary:**

The paper introduces a new benchmark to evaluate AI agents in open-ended machine learning research. Specifically, the paper first collects 201 research tasks from workshops of top machine learning conferences. LLMs are asked to complete end-to-end generation from a task into a full paper. The paper then divides the evaluation into four parts, including idea generation, proposal generation, experiment, and paper writing. The paper also uses LLM as a judge and predefined rubrics to evaluate results. The paper also develops a new agent to complete the task either in step-by-step or end-to-end. To ensure the quality of the MLR judge. The paper conducts an alignment evaluation between the LLM-judge and human evaluation. The paper also checks the key factors that affect research quality.

**Dataset Code Accessibility:**

Partly

**Dataset Code Comments:**

The paper provides code and data. However, some part of the benchmark is not well documented in the paper, as mentioned in the weakness.

**Ethical Considerations:**

No, there are no or only very minor ethics concerns

**Final Justification:**

Since the authors provided a very detailed rebuttal that addressed my questions, I decided to raise my score to 5.

**Limitations Weaknesses:**

1.  Although the task is useful, similar ideas have been proposed before, such as AIScientist [1], MLR-copilot [2].
2. Some parts are not clear. For step-by-step evaluation, it seems that the paper randomly selects a result as the input for the next step. However, the quality of such input is not clear.  What is the backbone model used to generate such output? A bad-quality output might also influence the downstream tasks. The definition of evaluation metrics for LLM-as-judge is also not shown in the paper. I also take a quick look at the task. It seems that the paper did not provide any specific dataset for the agent to work on. Does that mean the agent will automatically retrieve relevant data from the website? With a closer look at GitHub, it seems that such a dataset is not provided along with the task. What is the template for LLM-as-judge? What exactly is the input for step-by-step evaluation? What are the human evaluation details?
3. The baseline selection is confusing. Why does the end-to-end evaluation include three different models while the coding part only includes Claud? None of the open-source models is included in the end-to-end evaluation.


[1] Yamada, Y., Lange, R. T., Lu, C., Hu, S., Lu, C., Foerster, J., ... & Ha, D. (2025). The ai scientist-v2: Workshop-level automated scientific discovery via agentic tree search. arXiv preprint arXiv:2504.08066.

[2] Li, R., Patel, T., Wang, Q., & Du, X. (2024). Mlr-copilot: Autonomous machine learning research based on large language models agents. arXiv preprint arXiv:2408.14033.

**Strengths Contributions:**

1.  The proposed task is very useful and interesting. The paper includes a relatively large benchmark for automatic research discovery, about 201 research tasks.
2. The paper evaluates the agreement between LLM-as-judge and human evaluation. The agents are evaluated by both end-to-end and step-by-step.
3. The paper releases code and data. The visualization in the paper seems to be clear and easy to understand.

---

> ### Author Rebuttal · Authors · 2025-07-31
>
> We thank reviewer k5SZ for finding our task useful and interesting, and for recognizing the value of proposing a relatively large benchmark for automatic research discovery. We also appreciate the positive feedback on the clarity of our visualizations. Below, we address all the questions raised and hope our responses resolve the concerns.
>
> **W1: Although the task is useful, similar ideas have been proposed before, such as AIScientist [1], MLR-copilot [2].**
> We thank the reviewer for pointing out related agents. We would like to point out that our work is complementary rather than overlapping.
> - **Different focus**:
> While prior works like *AIScientist* and *MLR-Copilot* focus on **building research agents**, our work tackles a different but equally critical gap: how to **evaluate** such agents **systematically and scalably**.
>
> - **Our contributions**:
>   We propose **MLR-Bench**, the first benchmark tailored for open-ended ML research agent evaluation. It includes:
>   (1) a dataset of **201 real-world tasks** from NeurIPS, ICLR, and ICML workshops
>   (2) **MLR-Judge**: a structured, LLM-based evaluator **aligned with expert reviews**
>   (3) **MLR-Agent**: a modular scaffold for stepwise and end-to-end evaluation
>   These enable **fine-grained, comparative evaluation** across models and research stages—something prior works do not provide.
>
> - **Empirical findings on AIScientist V2**:
>   We also use MLR-Bench to evaluate existing agent such as AIScientist V2 (see Appendix A.2 in the supplementary materials):
>   - AIScientist V2 achieves relatively high score in *Novelty* and *Clarity* but underperforms in *Soundness* and *Significance*
>   - reflects a broader challenge we uncovered in multiple agents: lack of rigorous implementation, and frequently suffer from **result hallucination**
>
> - **Summary**:
>   MLR-Bench addresses a missing but critical component in this field: a **standardized, scalable, and reliable** evaluation pipeline. We believe it offers the community a valuable tool for diagnosing current limitations and guiding future progress. We encourage future agents to be evaluated under **MLR-Bench** for **fair comparison** and **continuous benchmarking**.
>
> ----------
> **Writing Clarity1: For step-by-step evaluation, it seems that the paper randomly selects a result as the input for the next step. However, the quality of such input is not clear. What is the backbone model used to generate such output? A bad-quality output might also influence the downstream tasks.**
>
> Thank you for the insightful comment! We would like to clarify that our step-by-step evaluation decomposes the research process into four stages: (1) idea generation, (2) proposal generation, (3) experimentation, and (4) paper writing. To evaluate each stage (e.g., experimentation) in isolation, we need to provide the model with a **fixed input from the previous stage** (e.g., a proposal teaching the model how to conduct experiments). We have considered the following four options as the input:
> 1. Each model uses its own output → unfair across models.
> 2. Use highest-scoring output → introduces bias.
> 3. **Randomly sample** from all outputs (**our choice**).
> 4. Manually write inputs → not scalable for 201 tasks × 4 stages.
>
> * Option (1) was problematic because different models would receive **different inputs**, making it **hard to isolate their performance** at each stage. As the reviewer noted, a poor-quality input could affect downstream results, and this would entangle the model’s current-stage ability with its performance in earlier stages.
>
> * Option (2) could **introduce bias**—if one model consistently performs well in an earlier stage (e.g., proposal generation), its output would be repeatedly selected as inputs for later stage, giving this model an unfair advantage—its scores in later stages might appear higher not because it is better at those stages, but simply because it receives inputs that are easier for it to process.
>
> * Option (4) was too **costly and not scalable** across 201 tasks and 4 stages, especially for the experimentation and writing stage. Additionally, in real-world usage, the input for a given task may be from humans or unknown AI models, and may not always be of especially high quality.
>
> * To balance consistency and fairness, we adopted Option (3): randomly sampling one output as the input for the next stage. This avoids overfitting to a single model’s output while keeping inputs consistent across models. As a result, the backbone model used to generate inputs varies across tasks and steps.
>
> We acknowledge that random sampling may include lower-quality outputs. However, since all models receive the same input for a given task, we believe the comparison remains fair—**any impact from input quality is shared equally** across models and does not affect the validity of relative performance measurements.
>
> ---
>
> **Writing Clarity2: What is the template for LLM-as-judge? What exactly is the input for step-by-step evaluation?**
>
> Thank you for the valuable feedback!
> - The **LLM-as-judge prompt templates** for each stage and end-to-end settings are described in detail in Appendix B.2.
>   - To form this template, we first define a set of **review rubrics** (e.g., Novelty, Clarity, Soundness) tailored to each research stage in **B.2.1**, then organize them with the model's output to form the judge input (see **B.2.2**).
>   - Table 25 shows an **example prompt** used to evaluate the writing.
> - Regarding the **step-by-step evaluation input**, we can also understand them through the **prompt template (Appendix B.1)**.
>  - Specifically, each stage receives a fixed input sampled from the outputs of the previous stage—for instance, the proposal generation stage takes a (task, idea) pair, and the paper writing stage takes a (task, idea, proposal, experiment log) tuple.
>  - To ensure fairness and avoid model-specific bias, we randomly sample from the generated outputs, as explained in the response above (Writing Clarity1).
>
> ---
>
> **Writing Clarity3: It seems that the paper did not provide any specific dataset for the agent to work on. Does that mean the agent will automatically retrieve relevant data from the website?**
>
> Yes! Since each research task may require a different dataset, we do not provide predefined datasets. Instead, we set up the environment to allow agents to **autonomously retrieve relevant data from online** sources such as Kaggle or Hugging Face, depending on the needs of the task.
>
> ---
>
> **Writing Clarity3: What are the human evaluation details?**
>
> Thank you for the feedback!
> - We recruited **10 reviewers** with experience at top-tier ML/AI venues.
> - Each reviewer was assigned several AI-generated papers aligned with their expertise.
> - Evaluation followed **OpenReview-style rubrics** (see **Appendix C and Figure 6/7 on the Google Doc Interface**):
>   - Rated on: Novelty, Soundness, Clarity, Significance, Overall Assessment
>   - Provided written feedback on strengths, weaknesses, and their confidence in the review.
>
> ---
>
> **Why does the end-to-end evaluation include three different models while the coding part only includes Claud? None of the open-source models is included in the end-to-end evaluation.**
>
> Thank you for the thoughtful question!
> * For the **coding agent**, at the time of submission (May 16), **Claude Code** was the only available coding agent that supported **autonomous web access** to **automatically complete full experiments**. Other agents we tested frequently failed on basic issues (e.g., import errors, environment setup), resulting in incomplete runs.
>   * We have since added **Codex** as an additional coding agent; its results are provided in our response to Reviewer xPbu.
> * For the **end-to-end evaluation**, the setting requires **multi-modal understanding and reasoning** since the experimentation step **produces several analytical figures**. Most open-source models at the time (e.g., **DeepSeek-R1**, **Ministral-8B** and **Qwen3-235B-A22B**) **did not support image input** so far, making them unsuitable.
>   * Currently, only **proprietary models** combine strong reasoning with image understanding, which is essential for faithfully simulating research workflows end-to-end.

---

> > ### Author Response · Authors · 2025-08-01
> > **Comments by Authors**
> >
> > To supplement **W3**, we have included end-to-end evaluation results for OpenAI **Codex** and Google **Gemini CLI**, which are now capable of executing experiments fully autonomously with network access. Since **Codex** currently only supports a `reasoning_effort of medium`, we use `o4-mini-medium` as its engine.
> >
> > 1. *Table A: End-to-end evaluation results over the ten tasks. We report the average ratings of two judges. We use `gemini-2.5-pro-preview-05-06` as the engine of Gemini-2.5-Pro-Preview  + Gemini CLI.*
> > | Model | Clarity | Novelty | Soundness | Significance | Overall |
> > | :--- | :--- | :--- | :--- | :--- | :--- |
> > | o4-mini-high + Codex | 6.45 ± 0.90 | 5.65 ± 0.60 | 2.90 ± 0.57 | 3.80 ± 0.65 | 3.10 ± 0.60 |
> > | o4-mini-high + Claude Code | 7.65 ± 0.32 | 6.90 ± 0.30 | 3.80 ± 0.60 | 5.05 ± 0.57 | 3.95 ± 0.71 |
> > | Gemini-2.5-Pro-Preview  + Gemini CLI | 8.30 ± 0.37 | 6.85 ± 0.34 | 4.15 ± 1.06 | 5.30 ± 0.91 | 4.60 ± 1.00  |
> > | Claude-3.7-Sonnet + Claude Code | 7.75 ± 0.34 | 7.10 ± 0.43 | 4.05 ± 1.11 | 5.50 ± 1.14 | 4.70 ± 1.22  |
> >
> > This table shows that, **Claude Code** outperforms **Codex** and **Gemini CLI** in end-to-end research, and `Claude-3.7-Sonnet` works better than `o4-mini-high`.
> >
> > 2. *Table B: End-to-end performance comparison of AI Scientist V2 and MLR-Agent on ten tasks, based on the average ratings of two judges. Both agents are powered by `o4-mini-high`. MLR-Agent utilizes Codex as its coding agent.*
> > | Agent Scaffold | Clarity | Novelty | Soundness | Significance | Overall |
> > | :--- | :--- | :--- | :--- | :--- | :--- |
> > | MLR-Agent | 6.45 ± 0.90 | 5.65 ± 0.60 | 2.90 ± 0.57 | 3.80 ± 0.65 | 3.10 ± 0.60 |
> > | AI Scientist V2 | 6.55 ± 0.94 | 6.70 ± 0.48 | 3.70 ± 1.29 | 4.85 ± 1.08 | 4.25 ± 1.25 |
> >
> > Using the same `o4-mini` model, **AI Scientist V2** outperforms **MLR-Agent**.

---

> > > ### Comment · Reviewer_k5SZ · 2025-08-01
> > >
> > > Thank you for your additional information. I have raised my score to 5.

---

> > > > ### Author Response · Authors · 2025-08-01
> > > >
> > > > Thank you for taking the time to review our response and for updating the score!

---

### Decision · Program_Chairs · 2025-09-18

**Decision:**

Accept (poster)

**Comment:**

This paper introduced MLR-Bench, a benchmark for AI agents in ML research. 201 tasks are extracted from top ML conference workshops, an automated LLM as a judge framework is provided, and a new agent to complete tasks is provided.

Reviewer k5SZ values the components. Two relevant papers are provided. The authors argue that this paper is more focused on evaluation rather than building agents. The paper provided insights into AI agent paper that is discussed here. Additional simulation tables on frontier AI agents are provided during the discussion.
Reviewer 9DJw also cited two relevant papers. The reviewer values the fact that this paper involves human evaluation and is satisfied with the author's response.
Reviewer xPbu values the AI agents for scientific discovery and system architecture. The limitation is that there is only a single coding agent.
Reviewer ESS7 discusses seven limitations, but still this paper is acceptable, though there are limitations on ("address the core architectural problems").

In summary, since all reviewers consider this paper above the threshold, I recommend a publication in the track.